# GTA: GRAPH THEORY AGENT AND BENCHMARK FOR ALGORITHMIC GRAPH REASONING WITH LLMS

## ABSTRACT

Large Language Models (LLMs) are increasingly being applied to tasks involving structured data such as graphs, yet their capacity for complex algorithmic reasoning over graph-structured inputs remains underexplored. Existing benchmarks often involve overly simplistic tasks defined on small graphs, or focus primarily on code generation rather than direct reasoning over graph structures, or rely on a single input format, which limits a comprehensive evaluation of LLMs' capabilities in graph reasoning. To address this gap, we introduce **Graph Theory Bench (GT Bench)**, a challenging new benchmark featuring 44 diverse graph problem types across over 100,000 instances with varied input representations (natural language, structured language, adjacency list, adjacency matrix). GT Bench is specifically designed to evaluate the ability of LLMs to perform multi-step algorithmic reasoning on graph-structured tasks. Beyond benchmarking the performance of various LLM on graph reasoning tasks, our experiments reveal a critical insight: the effectiveness of LLMs is closely tied to the choice of input graph representation, and this dependency is further influenced by intrinsic graph properties such as density, size, and topology. Based on these findings, we propose the **Graph Theory Agent (GTA)**, a novel framework that enhances LLM graph reasoning by employing an adaptive input representation selector and decomposing the algorithmic solution into manageable sub-steps. Experiments demonstrate that GTA significantly improves the ability of LLMs to solve complex graph problems.

## 1 INTRODUCTION

Large Language Models (LLMs) have demonstrated remarkable capabilities across various domains, achieving significant success in areas such as social science applications (Huang et al., 2024; Xu et al., 2025), medical advancements (Zhou et al., 2024; Tian et al., 2024; Liu et al., 2023; Chen et al., 2025), and robotics domain Song et al. (2024). With the expanding range of LLM applications, these models are increasingly expected to handle and reason about intricate structured data, with graph-structured representations playing a particularly prominent role. This necessity spans diverse contexts, from analyzing intricate social networks Wang et al. (2025a) and knowledge graphs Wu et al. (2024b) to understanding molecular Pan (2023) interactions and optimizing logistical systems. Effectively tackling these challenges demands sophisticated LLM abilities in interpreting graph structures and performing multi-step reasoning.

However, existing benchmarks designed to evaluate LLMs on graph-related tasks often fall short in assessing these crucial reasoning capabilities, primarily following two distinct paradigms. First, some benchmarks focus on code generation for graph problems (Wu et al., 2024a; Yuan et al., 2024), where models are evaluated on their ability to produce code solutions without processing the input graph or performing any actual reasoning. Second, other benchmarks do require models to reason over graph inputs, but the tasks are typically overly simplistic (e.g., identifying immediate neighbors) or restricted to small-scale graphs, which limits their ability to assess multi-step, algorithmic reasoning (Jin et al., 2024; Li et al., 2023). Moreover, these benchmarks often rely on a single, fixed input representation, further constraining the evaluation of a model's adaptability and robustness across different graph encodings. As a result, current evaluations largely overlook the essential capability of LLMs to perform complex, structured reasoning over graphs of non-trivial scale that are still amenable to language-based reasoning.

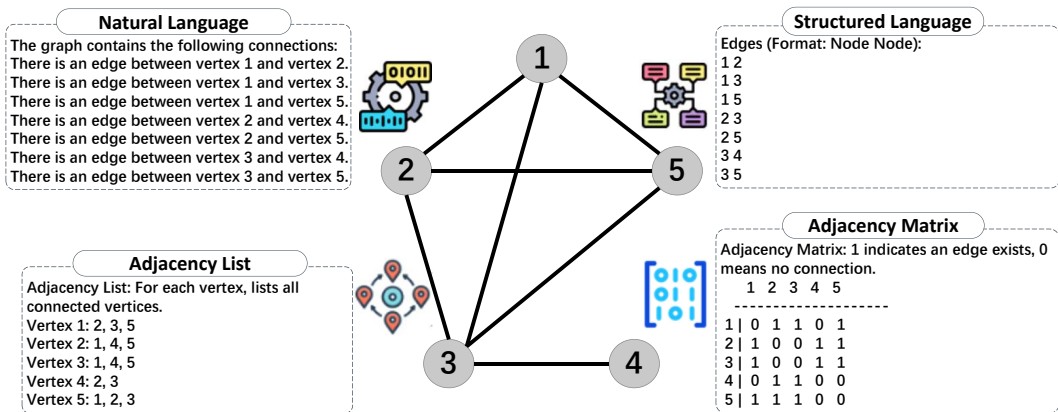

Figure 1: Example of an undirected graph represented using the four input modalities evaluated in this work: Natural Language, Structured Language, Adjacency List, and Adjacency Matrix.

To address this deficiency, we introduce **Graph Theory Bench (GT Bench)**, a novel benchmark specifically curated to assess the multi-step reasoning capabilities of LLMs on graph-theoretic problems with practical computational and structural complexity. GT Bench is fully open-source, with data, generators, and evaluation scripts released. GT Bench comprises a diverse collection of 44 distinct graph problem scenarios, with over 600 instances generated for each scenario, totaling more than 100,000 problem instances overall. The dataset is stratified into *easy* and *hard* subsets, covering a wide spectrum of tasks ranging from basic connectivity checks to complex problems like minimum-cost maximum-flow (MCMF). Each problem instance is presented across four different input modalities (natural language, structured language, adjacency lists, and adjacency matrices) to rigorously test model robustness and adaptability, and to explore the impact of graph representation on reasoning performance. A parameterized generation pipeline supports flexible control of graph families (e.g., directed/undirected, weighted/unweighted, sparse/dense, tree-structured) and sizes $(|V|, |E|)$, enabling reproducible extensions. We provide a comprehensive, automated generation pipeline that ensures reproducibility and facilitates future extensions; in the released corpus, node and edge counts are calibrated to contemporary LLM context windows—large enough to require genuine multi-step reasoning yet compact enough to fit reliably within prompts. Unlike previous efforts focusing on code generation or rudimentary graph queries, GT Bench deliberately targets challenging reasoning beyond simple graph understanding: its tasks are constructed to necessitate algorithmic thinking and combinatorial reasoning (rather than surface-level parsing), thereby providing a more targeted evaluation of complex graph problem-solving abilities in LLMs.

Our experiments on GT Bench uncovered a critical dependency between the input graph representation and LLM task performance, which varied significantly with graph structure. For instance, adjacency matrix representations yielded superior results on dense graphs, natural language descriptions proved more effective for sparse graphs, and structured language formats were particularly beneficial for tree structures. This finding highlights that the choice of representation is crucial for effective graph reasoning. Based on these findings, we propose the **Graph Theory Agent (GTA)** to enhance the graph reasoning capabilities of LLMs. GTA is designed as a multi-component agent that first employs an *Input Representation Selector*—implementable via heuristics or a fine-tuned LLM—to choose the most suitable graph format for the given problem and graph structure. Subsequently, an *Algorithm Generator* produces a high-level algorithmic plan, which is then broken down into manageable sub-steps by an *Algorithm Decomposer* (a fine-tuned LLM) for sequential processing by an *Executor*. This structured, adaptive approach aims to leverage the strengths of different representations and guide the model through complex reasoning processes, thereby significantly improving performance on challenging graph-theoretic tasks.

In summary, our main contributions are threefold: (1) We introduce GT Bench, a comprehensive and challenging benchmark explicitly designed to evaluate the multi-step algorithmic reasoning capabilities of LLMs on graph-theoretic problems with realistic structural and computational complexity. (2) We provide extensive empirical analysis using GT Bench, revealing key insights into current LLM limitations and the crucial role of input representation choice. (3) We propose the GTA, a

novel agent framework that adaptively selects representations and decomposes problem-solving steps, demonstrably enhancing the graph reasoning performance of LLMs. These contributions advance the understanding and evaluation of LLM reasoning on structured graphs and highlight avenues for improvement.

## 2 GT BENCH: GRAPH THEORY BENCHMARK

To rigorously evaluate LLM algorithmic reasoning on graphs, we constructed GT Bench to probe multi-step inference across diverse graph structures and graph-theoretic problems. Detailed task definitions and ground-truth algorithms are provided in Appendix A.

### 2.1 GRAPH CHARACTERISTICS

The graphs within GT Bench are primarily categorized by edge density into sparse graphs, where the number of edges $E$ scales linearly with the number of vertices $V$ (i.e., $E = O(V)$), and dense graphs, where $E$ approaches the quadratic maximum ($E = O(V^2)$). A third significant category comprises trees, defined as connected graphs satisfying $E = V - 1$. Depending on specific task requirements, graphs are generated with varying properties, such as weighted or unweighted edges, and may be connected or disconnected. Specific topological structures like star graphs are also implicitly generated within these categories.

### 2.2 INPUT REPRESENTATIONS

A distinctive feature of GT Bench is its provision of each graph problem instance in four distinct input modalities, facilitating a comprehensive study of model robustness and the impact of representation choice on reasoning performance. These modalities encompass: **Natural Language (NL)**, a descriptive prose format detailing vertices, edges, and weights where applicable; **Structured Language (SL)**, which employs a more templated approach using keywords and indentation to articulate graph structure; the **Adjacency Matrix (AM)**, a matrix-based representation where entries denote edge existence or their associated weights; and the **Adjacency List (AL)**, which lists the neighbors and corresponding weights for each vertex. Figure 1 offers a visual comparison of these four representations applied to an exemplary small graph.

### 2.3 TASK COVERAGE AND DATASET COMPOSITION

GT Bench encompasses 44 distinct types of classical graph-theoretic problems, designed to cover a broad spectrum of complexities. These range from fundamental tasks such as connectivity checks to advanced optimization challenges like MCMF. For each task, the graphs in the dataset are of a medium scale, intended to pose a significant yet not excessive challenge to LLMs. Table 1 details the node counts and instance numbers for each problem type. Collectively, the dataset comprises over 100,000 instances, which are further stratified into *easy* (GT-E) and *hard* (GT-H) subsets to accommodate varying levels of difficulty, where details can be found in Appendix A.

The generation of the entire dataset, including graph structures, problem formulations, and solution verifications, is fully automated through algorithmic procedures. This generation algorithm is also flexible, allowing for the arbitrary adjustment of graph scales to produce new data instances. Ground truth solutions are systematically derived using canonical algorithms specific to each task, thereby ensuring correctness and eliminating the need for manual annotation or labeling. Both the GT Bench dataset and the one-click algorithm for generating data have been made open-source.

### 2.4 REPRESENTATION SENSITIVITY

Our experiments (detailed in subsection 4.2) underscore that input representation choice critically influences LLM performance on GT Bench tasks. As shown in Table 2, no single format is universally optimal—the best choice depends on the specific problem and underlying graph structure (e.g., density, topology). This sensitivity is pivotal for effective LLM-based graph reasoning. A closer look at the top-performing formats reveals distinct patterns that offer insights into format-task alignment.

Table 1: Overview of Tasks and Statistics in the GT Bench Dataset.

| Task | Description | Nodes | Instances |
|---|---|---|---|
| Connectivity | Determine whether two vertices are connected | 15–40 | 4800 |
| Bipartiteness Check | Determine whether the graph is bipartite | 15–40 | 4800 |
| Minimum Cycle Length | Find the length of the smallest cycle in the graph | 15–40 | 4800 |
| Maximum Clique Size | Compute the size of the largest clique | 15–40 | 4800 |
| Maximum Independent Set | Compute the size of the largest independent set | 15–40 | 4800 |
| Eulerian Path | Determine whether the graph contains an Eulerian path | 15–40 | 4800 |
| Eulerian Circuit | Determine whether the graph contains an Eulerian circuit | 15–40 | 4800 |
| Hamiltonian Path | Determine whether the graph contains a Hamiltonian path | 15–40 | 4800 |
| Hamiltonian Circuit | Determine whether the graph contains a Hamiltonian circuit | 15–40 | 4800 |
| Biconnected Components | Count the number of biconnected components | 11–20 | 4800 |
| Bridge Count | Count the number of bridges (cut edges) | 11–20 | 4800 |
| Triangle Count | Count the number of triangles (3-node cycles) | 11–13 | 4800 |
| Cycle Count | Count the total number of cycles in the graph | 6–10 | 4800 |
| Spanning Tree Count | Count the number of possible spanning trees | 6–10 | 4800 |
| Shortest Path Length | Compute the shortest path between two nodes | 14–20 | 4800 |
| Minimum Spanning Tree | Compute the total weight of the minimum spanning tree | 14–20 | 4800 |
| Second Minimum Spanning Tree | Compute the weight of the second-best minimum spanning tree | 11–20 | 4800 |
| Tree Diameter | Compute the diameter (longest shortest path) of the tree | 29–31 | 2400 |
| Tree Centroid | Find the centroid node with the smallest index | 29–60 | 2400 |
| Lowest Common Ancestor | Find the lowest common ancestor (root at node 1) | 29–60 | 2400 |
| Tree Max Independent Set | Compute the size of the maximum independent set in a tree | 29–60 | 2400 |
| Maximum Flow | Compute the maximum flow from source to sink | 11–20 | 4800 |
| Minimum Cut | Compute the capacity of the minimum cut (source to sink) | 11–20 | 4800 |
| Min-Cost Max-Flow | Compute the minimum-cost maximum flow | 9–15 | 4800 |

**NL** representations demonstrated utility for specific tasks on sparse graphs. NL's descriptive prose can facilitate a qualitative, high-level understanding of simpler graph topologies, potentially leveraging the LLM's linguistic strengths when extensive structural parsing is less critical.

**SL** representations excelled for tasks on tree structures and several sparse graph problems. Its templated, hierarchical format naturally mirrors tree relationships, aiding algorithms reliant on structured traversal. For sparse graphs, SL provides organized enumeration of connections, offering clarity over NL without the density issues of AM for largely empty structures.

**AM** representations proved superior for tasks on dense graphs. AM's matrix format provides a comprehensive and direct view of all potential pairwise connections, distinctly showing both present and absent edges. This complete structure can aid LLMs in discerning global graph properties and patterns, particularly in dense graphs where numerous relationships are simultaneously relevant.

**AL** formats performed strongly, especially for tasks centered around pathfinding, local neighborhood exploration, and many sparse graph problems, along with select dense graph tasks. AL explicitly lists existing edges, rendering it efficient for algorithms that iterate through neighbors (e.g., foundational search algorithms). This efficiency is particularly pronounced in sparse graphs. For certain dense graph tasks as well, algorithms benefit from the direct edge iteration AL effectively provides.

These empirical findings highlight LLMs' considerable sensitivity to the presentation modality of graph information. This observed variability underscores the pivotal role of representation choice and directly motivates the adaptive *Input Representation Selector* component within our proposed **GTA** framework (section 3). GTA aims to leverage these insights by dynamically selecting the most suitable format, thereby enhancing LLM performance on complex graph problems.

## 3 GRAPH THEORY AGENT FRAMEWORK

Our experiments with GT Bench (introduced later in section 4) highlight critical bottlenecks for current LLMs attempting complex graph reasoning: inconsistent performance across models and a strong dependency on how the graph data is presented (subsection 2.4). These findings motivate the development of specialized approaches. We introduce the **GTA**, a framework engineered to improve algorithmic graph reasoning in LLMs through two key strategies: adaptive selection of input

Table 2: Best-performing input Representation format (Best Repr.) per task and graph type.

| Task & Graph Type | Best Repr. | Task & Graph Type | Best Repr. |
|---|---|---|---|
| Biconnected Components (Dense) | AM | Maximum Independent Set (Dense) | AM |
| Biconnected Components (Sparse) | SL | Maximum Independent Set (Sparse) | AM |
| Bipartite (Dense) | AM | Min Cost Max Flow (Dense) | AM |
| Bipartite (Sparse) | AL | Min Cost Max Flow (Sparse) | AL |
| Bridge Count (Dense) | AL | Minimum Cut (Dense) | AM |
| Bridge Count (Sparse) | NL | Minimum Cut (Sparse) | AL |
| Connectivity (Dense) | AM | Minimum Cycle (Dense) | AM |
| Connectivity (Sparse) | AL | Minimum Cycle (Sparse) | NL |
| Cycle Count (Dense) | AL | Minimum Spanning Tree (Dense) | AL |
| Cycle Count (Sparse) | AM | Minimum Spanning Tree (Sparse) | SL |
| Eulerian Circuit (Dense) | AM | Second MST (Dense) | SL |
| Eulerian Circuit (Sparse) | SL | Second MST (Sparse) | SL |
| Eulerian Path (Dense) | AM | Shortest Path (Dense) | AL |
| Eulerian Path (Sparse) | AM | Shortest Path (Sparse) | AL |
| Hamiltonian Circuit (Dense) | AL | Spanning Tree Count (Dense) | AM |
| Hamiltonian Circuit (Sparse) | AL | Spanning Tree Count (Sparse) | AL |
| Hamiltonian Path (Dense) | AL | Tree Centroid (Tree) | SL |
| Hamiltonian Path (Sparse) | AL | Tree Diameter (Tree) | SL |
| Maximum Clique (Dense) | AM | Tree LCA (Tree) | SL |
| Maximum Clique (Sparse) | NL | Tree Max Independent Set (Tree) | SL |
| Maximum Flow (Dense) | AM | Triangle Count (Dense) | AM |
| Maximum Flow (Sparse) | AL | Triangle Count (Sparse) | NL |

representations and systematic decomposition of the problem-solving process. GTA tackles a graph problem instance $P = (G, T)$ (where $G$ is the graph and $T$ the task) via a structured pipeline:

$$P \xrightarrow{\text{Selector}} P' \xrightarrow{\text{Generator}} \mathcal{A} \xrightarrow{\text{Decomposer}} \{s_1, ..., s_k\} \xrightarrow{\text{Executor}} \text{Solution} \qquad (1)$$

This process initiates with the **Input Representation Selector** choosing an optimal format $R^*$ for graph $G$ concerning task $T$, yielding $P' = (G_{R^*}, T)$. Subsequently, the **Algorithm Generator** formulates a high-level strategy $\mathcal{A}$. This strategy is then refined by the **Algorithm Decomposer** into a sequence of manageable sub-steps $\{s_1, ..., s_k\}$. Finally, an **Executor** LLM processes these sub-steps to compute the solution.

### 3.1 INPUT REPRESENTATION SELECTOR

The Input Representation Selector is the initial component in the GTA pipeline. Its crucial role is to dynamically choose the most effective format from the available options $\mathcal{R} = \{\text{NL, SL, AM, AL}\}$ to present the input graph $G$ for a given task $T$. The objective is to select the representation $R^*$ that maximizes the likelihood of successful downstream reasoning by the Executor LLM. This selection can be formalized as $\mathcal{S} : (G, T) \mapsto R^*$, where $R^* \in \mathcal{R}$ is determined based on features of $G$ (e.g., size, density) and the requirements of $T$. We explore two implementations for this adaptive selector:

**1. Heuristic-Based Selector:** This approach employs a rule-based function derived from our empirical findings (Table 2). It maps observable graph and task characteristics to the representation that demonstrated the strongest average performance. Its primary advantage is computational efficiency.

**2. Learned Selector:** Alternatively, this selector is a dedicated language model, fine-tuned for representation selection. This allows optimization for a *specific* downstream Executor LLM, trained to predict the representation most likely to yield successful task completion by that target model using historical data of problem instances, representations, and outcomes. Henceforth, unless otherwise specified, references to GTA will imply the use of the Learned Selector.

By adaptively tailoring the input format, the Input Representation Selector directly addresses the representation sensitivity challenge highlighted by GT Bench, ensuring subsequent components receive graph data in a structure conducive to effective reasoning.

## 3.2 ALGORITHM GENERATOR

Following the selection of an optimal representation $R^*$, the **Algorithm Generator** formulates a *high-level* strategic plan $\mathcal{A}$. This component, an off-the-shelf LLM, is prompted with the task description $T$ and the graph $G_{R^*}$ in its chosen format. Its role is conceptual: to establish the overarching algorithmic approach and outline key reasoning phases. It explicitly avoids generating detailed instructions or executable code. The resultant plan $\mathcal{A}$ serves as a strategic blueprint, ensuring a logically sound methodology and providing high-level guidance for the subsequent Algorithm Decomposer. This component operates without task-specific fine-tuning for its abstract plan generation stage.

## 3.3 ALGORITHM DECOMPOSER

Bridging the gap between the high-level plan $\mathcal{A}$ generated previously and the final execution phase, the **Algorithm Decomposer** refines the abstract strategy into a concise sequence of concrete, manageable sub-steps $\{s_1, s_2, ..., s_k\}$. These sub-steps are specifically crafted for sequential processing by the LLM Executor. By breaking down the larger problem, this decomposition aims to mitigate the cognitive complexity for the Executor, guiding its focus incrementally through the algorithmic logic. The transformation performed by the decomposer can be represented as:

$$\text{Decomposer} : (\mathcal{A}, G_{R^*}) \mapsto \{s_1, s_2, ..., s_k\} \tag{2}$$

Each resulting sub-step $s_i$ constitutes a focused natural language instruction that directs the Executor to perform a specific part of the overall algorithm. To effectively translate conceptual plans into actionable steps, the Algorithm Decomposer is implemented as a dedicated LLM. The specific fine-tuning process designed to optimize its decomposition capabilities is detailed in subsection 3.4.

## 3.4 TRAINING AND EXECUTION WORKFLOW

The GTA framework's components are trained sequentially. The **Heuristic-Based Selector** is a parameter-free function based on empirical results, and the **Algorithm Generator** employs a base LLM without specific fine-tuning for its role. The core training focuses on the **Algorithm Decomposer** and subsequently the **Learned Selector**.

**Algorithm Decomposer Training:** The Algorithm Decomposer LLM is trained in two stages, using the Heuristic-Based Selector for input graph representation ($G_{R^*_{\text{heuristic}}}$). *1. Supervised Fine-Tuning (SFT):* To establish foundational decomposition skills, the Algorithm Decomposer LLM is initially fine-tuned on expert-like decompositions $D_{\text{expert}}$ (obtained via distillation from a stronger model) for given plans $\mathcal{A}$ and graphs $G_{R^*_{\text{heuristic}}}$. This stage minimizes the SFT loss:

$$\mathcal{L}_{\text{SFT-Decomp}} = -\mathbb{E}_{(\mathcal{A}, G_{R^*_{\text{heuristic}}}), D_{\text{expert}}} \left[ \log P_{\text{Decomp}}(D_{\text{expert}} | \mathcal{A}, G_{R^*_{\text{heuristic}}}) \right], \tag{3}$$

where $P_{\text{Decomp}}$ denotes the probability assigned by the Algorithm Decomposer LLM. *2. Direct Preference Optimization (DPO):* The SFT-tuned model is then further refined. Multiple decomposition candidates $\{D_j\}$ are sampled for each problem. After execution by the Executor, these are labeled as "winning" ($D_w$, leading to a correct solution) or "losing" ($D_l$, leading to an incorrect solution). DPO then optimizes the Algorithm Decomposer LLM using these preference pairs $(D_w, D_l)$ to maximize the likelihood of generating effective decompositions for the Executor. This directly links the Decomposer's output to successful problem execution.

**Learned Selector Training:** With the Algorithm Decomposer trained and fixed, the Learned Selector LLM is trained. For each problem instance $P = (G, T)$, the complete GTA pipeline (using the Algorithm Generator, Algorithm Decomposer, and Executor) is executed with each of the four input representations $R \in \{\text{NL, SL, AM, AL}\}$. This yields preferred representations $R_w$ (leading to correct solutions) and dispreferred ones $R_l$. The Learned Selector is then fine-tuned via DPO using these preference pairs $(R_w, R_l)$ for each problem $P$. This optimizes the Selector to choose the representation that maximizes the end-to-end success probability of the GTA pipeline.

**Execution Phase:** During inference, the chosen Selector (either Heuristic-Based or Learned Selector) determines the input representation $R^*$. The Algorithm Generator formulates the high-level plan $\mathcal{A}$. The Algorithm Decomposer breaks $\mathcal{A}$ into sub-steps $\{s_i\}$, which are then processed by the base LLM acting as the Executor to derive the final solution.

## 4 EXPERIMENTS

### 4.1 EXPERIMENTAL SETUP

**Datasets.** Our primary evaluations are conducted on the newly introduced GT Bench. The generalization of our method is additionally validated on the Graph Understanding subset of GraCoRe (Yuan et al., 2024) and the complete NLGraph (Wang et al., 2023) dataset. Instances from GraCoRe and NLGraph are randomly converted to one of the four input modalities (NL, SL, AL, AM) used in GT Bench.

**Models.** We evaluate a diverse set of eight LLMs on GT Bench. These include three proprietary models: `GPT-4o-mini` (OpenAI, 2024), `GPT-4o` (Hurst et al., 2024), `o3-mini` (OpenAI, 2025); and five open-source models: `Llama-3.1-8B` (Meta, 2024b), `Llama-3.3-70B` (Meta, 2024a), `Phi-4` (Abdin et al., 2024), `QwQ-32B` (Team, 2025) and `DeepSeek-R1` (Guo et al., 2025).

**Baselines.** Across all datasets, we compare our proposed GTA against several established methods. These encompass prompting-based techniques: Vanilla prompting, Chain-of-Thought (CoT) (Wei et al., 2022), LLM-Debate (Du et al., 2023), and Self-Refine (Madaan et al., 2023); as well as automated agent frameworks: ADAS (Hu et al., 2024a) , AFlow (Zhang et al., 2024a) and MaAS (Zhang et al., 2025). To ensure a fair comparison, all baseline methods also utilize the `Phi-4` model as their executor LLM.

**Implementation Details.** The GTA is trained solely on GT Bench, with its generalization performance evaluated on GraCoRe and NLGraph. We utilize the `Phi-4` model for GTA's Learned Selector, Algorithm Generator, Algorithm Decomposer, and Executor components to balance performance and computational cost. Expert demonstrations for training the Algorithm Decomposer are distilled from `GPT-4o`. A consistent temperature of 0.01 is used for all model inferences to ensure stability. Further details regarding the experimental configurations are provided in Appendix C. Extended analyses are in Appendix D—Easy/Hard splits, representation effects, scalability, cost, executor swaps, and SFT–DPO dynamics.

Table 3: Accuracy (%) of tested models on GT Bench across different tasks and input representations.

| Task | Graph Type | NL | SL | AM | AL | Average | Range |
|------|-----------|------|------|------|------|---------|-------|
| Bipartite Check | Sparse | **89.7** | 83.8 | 76.1 | 76.9 | 81.6 | 13.6 |
| Tree Centroid | Tree | 56.4 | **60.7** | 42.7 | 51.3 | 52.8 | 18.0 |
| Maximum Clique | Dense | 24.8 | 22.2 | **38.5** | 35.0 | 30.1 | 16.3 |
| Hamiltonian Circuit | Dense | 81.2 | 81.2 | 82.9 | **88.0** | 83.3 | 6.8 |

### 4.2 REPRESENTATION SENSITIVITY ON GT BENCH

Our analysis of LLM performance on GT Bench reveals critical insights into their interaction with different graph input representations.

**Different models exhibit distinct preferences for input representations.** As shown in Table 4, no single input format is universally optimal. For example, `llama-3.1-8B` favors AM (29.4%), while `phi-4` performs best with SL (44.9%). Larger model like `gpt-4o` achieve peak accuracy with AL (47.6%), whereas `deepseek-r1` prefers AM (82.5%). This model-dependent variability underscores the challenge of representation selection and highlights the rationale behind adaptive approaches.

Table 4: Accuracy (%) of different models on GT Bench under four input representations.

| Model | NL | SL | AM | AL |
|-------|------|------|------|------|
| Llama-3.1-8B | 29.0 | 27.4 | **29.4** | 28.5 |
| Phi-4 | 43.7 | **44.9** | 38.8 | 42.7 |
| GPT-4o-mini | 36.2 | 33.4 | 37.8 | **40.7** |
| Llama-3.3-70B | **44.9** | 41.1 | 40.7 | 44.1 |
| GPT-4o | 45.5 | 43.7 | 45.3 | **47.6** |
| QwQ-32B | 63.1 | 63.1 | 62.1 | **71.3** |
| DeepSeek-R1 | 80.9 | 80.6 | **82.5** | 80.9 |
| o3-mini | 87.4 | 89.9 | 89.0 | **90.2** |

**The optimal input representation is highly dependent on the specific task and graph structure.** Table 3 illustrates this dependency with selected tasks from GT Bench. For instance, NL representation (89.7%) is most effective for Bipartite Check on sparse graphs, whereas AM (38.5%) is superior for Maximum Clique on dense graphs. Tree-centric tasks like Tree Centroid benefit

from SL (60.7%), and path-oriented problems such as Hamiltonian Circuit on dense graphs achieve highest accuracy with AL (88.0%). These findings, detailed comprehensively in Appendix D, directly informed the design of the heuristic input selector within our GTA framework.

## 4.3 COMPARATIVE PERFORMANCE OF GTA

To evaluate the effectiveness of our proposed GTA, we compare its performance against several established prompting strategies and automated agent frameworks. The average accuracy across GT Bench (Easy and Hard subsets), GraCoRe, and NLGraph is presented in Table 5.

The results indicate that while most advanced prompting techniques and agent frameworks provide varying degrees of improvement over vanilla prompting for graph reasoning tasks, our GTA consistently achieves superior performance across all evaluated benchmarks. Specifically, on GT Bench Easy (GT-E) and Hard (GT-H) subsets, GTA obtains accuracies of 69.1% and 41.5%, respectively, outperforming the next best method, MaAS (63.2% on GT-E and 37.6% on GT-H) and AFlow (62.0% on GT-E and 37.2% on GT-H), by a significant margin.

GTA is trained solely on the GT Bench dataset. Despite this, it achieves high accuracy on GraCoRe (80.2%) and NLGraph (89.4%), two benchmarks with task distributions that are not direct subsets of GT Bench. These results indi-

Table 5: Average accuracy (%) of different methods on multiple graph reasoning benchmarks. GT-E = GT Bench (Easy), GT-H = GT Bench (Hard), GCR = GraCoRe, NLG = NLGraph.

| Method | GT-E | GT-H | GCR | NLG |
|---|---|---|---|---|
| Vanilla | 53.5 | 33.0 | 66.8 | 80.2 |
| CoT | 52.1 | 34.2 | 67.2 | 81.0 |
| LLM-Debate | 58.9 | 35.5 | 70.3 | 84.4 |
| Self-Refine | 56.8 | 34.6 | 68.0 | 83.1 |
| GraphTeam | 50.1 | 27.2 | 51.6 | 63.2 |
| ADAS | 52.8 | 32.7 | 67.5 | 80.5 |
| AFlow | 62.0 | 37.2 | 75.4 | 86.2 |
| MaAS | 63.2 | 37.6 | 78.8 | 86.0 |
| **GTA** | **69.1** | **41.5** | **80.2** | **89.4** |

cate that GTA possesses strong generalization capabilities across diverse graph reasoning tasks. The effectiveness of its adaptive input representation selection and structured problem decomposition contributes to its ability to handle related yet distinct reasoning challenges. The consistent performance gains highlight the advantages of GTA's design in improving the capacity of LLM to solve complex algorithmic graph problems.

## 4.4 ABLATION STUDY OF GTA COMPONENTS

To verify the contribution of its core modules, we conducted ablation studies on our GTA framework. We evaluated variants by removing or simplifying key components, comparing them against our complete **GTA** (with the Learned Selector) and a **Vanilla** baseline (Phi-4 with vanilla prompting). The variations include: **GTA-NoSel**: GTA without any Input Representation Selector (fixed default representation). **GTA-NoDec**: GTA without the Algorithm Generator and Decomposer (direct execution). **GTA-HeuSel**: GTA using the Heuristic-Based Selector instead of the Learned Selector. Performance is reported in Table 6.

The results clearly demonstrate the importance of each component. Comparing GTA-NoSel to GTA, the absence of any input representation selection leads to a significant performance drop across all datasets (e.g., from 69.1% to 58.2% on GT-E). This underscores the critical impact of adapting the input format to the problem and model. The GTA-HeuSel configuration, employing a heuristic approach for selection, shows substantial improvement over GTA-NoSel. However, it does not match the performance of GTA with its Learned Selector, highlighting the added value derived from a data-driven, fine-tuned selection

Table 6: Accuracy (%) on GT Bench, GraCoRe (GCR), and NLGraph (NLG). Configurations: Vanilla (Vanilla Phi-4), GTA-NoSel (w/o Selector), GTA-HeuSel (w/ Heuristic Selector), GTA-NoDec (w/o Generator & Decomposer), **GTA** (Our proposed method).

| Config. | GT-E | GT-H | GCR | NLG |
|---|---|---|---|---|
| Vanilla | 53.5 | 33.0 | 66.8 | 80.2 |
| GTA-NoSel | 58.2 | 36.1 | 71.6 | 83.5 |
| GTA-NoDec | 62.1 | 37.7 | 75.3 | 85.0 |
| GTA-HeuSel | 64.5 | 38.8 | 76.5 | 85.8 |
| **GTA** | **69.1** | **41.5** | **80.2** | **89.4** |

mechanism that can better adapt to the nuances of the executor LLM and task types.

Furthermore, the GTA-NoDec variant, which omits the explicit planning and decomposition stages, also performs notably worse than GTA (e.g., 62.1% vs. 69.1% on GT-E). This finding validates the benefit of guiding the LLM through multi-step reasoning via a structured algorithmic breakdown, rather than expecting it to solve complex problems monolithically. Consistently, our proposed GTA framework, integrating all its designed components, achieves the highest accuracy. This confirms that the synergistic combination of adaptive input representation selection and structured algorithmic decomposition is pivotal to its superior performance on complex graph reasoning tasks.

## 4.5 EXPANDED EXPERIMENTAL ANALYSES

To complement the main results, we provide additional experiments in the Appendix that probe GT-Bench and GTA from multiple aspects. These analyses clarify dataset positioning, ensure fair baseline comparisons, expose detailed per-task and per-representation outcomes, and evaluate scalability, efficiency, and robustness. Together, they strengthen our claims and offer a broader view of how GT-Bench and GTA jointly advance language-based graph reasoning.

**(1) Benchmark positioning.** In subsection B.1, we revisit existing datasets and show how GT-Bench fills a gap in evaluating language-based algorithmic reasoning. Prior work often emphasizes coding, semantic QA, or domain-specific settings, while GT-Bench expands scale, task diversity, and input modalities to directly test reasoning over graph structure.

**(2) Methods for graph-theoretic problems.** As discussed in subsection B.2 and subsection C.2, many existing systems either target different tasks, rely on heavy multi-agent schemes, or bypass reasoning via code execution. GTA instead keeps the executor frozen and improves performance through representation selection and lightweight decomposition. For fair comparison, all baselines are standardized on `Phi-4`, with details provided in the appendix.

**(3) Detailed results and input–representation sensitivity.** subsection D.1 shows accuracy drops sharply on the Hard subset, highlighting the difficulty of complex reasoning. In subsection D.2, we further observe strong variability across input formats: NL favors sparse tasks, SL works well for trees, AM excels on dense global properties, and AL is best for paths and flows. This motivates GTA's adaptive selector.

**(4) Scalability with increasing graph size.** From subsection D.3, small models degrade quickly as graphs grow, while GTA maintains stable accuracy on simpler tasks and degrades more gracefully on harder ones, consistently outperforming its base executor.

**(5) Inference cost analysis.** subsection D.4 shows that multi-round frameworks like Debate or Self-Refine are prohibitively costly, whereas GTA achieves notable gains with only modest overhead, close to lightweight agents like MaAS.

**(6) GTA-specific ablations and extensions.** As shown in subsection D.5, GTA improves performance across diverse executors, even for strong models. Training analysis in subsection D.6 confirms that SFT ensures broad coverage, while DPO yields consistent improvements without introducing representation bias.

## 5 CONCLUSION

This work advances LLM capabilities in complex, multi-step algorithmic reasoning on graph-structured data—an important yet underexplored challenge. We introduce Graph Theory Bench (GT Bench), a comprehensive benchmark for evaluating LLM reasoning across diverse graph problems and input formats. Our results show that LLM performance is highly sensitive to input representation, which varies with graph structure. To address this, we propose the Graph Theory Agent (GTA), a novel framework that improves LLM reasoning by adaptively selecting input formats and decomposing solutions into manageable sub-steps. Experiments demonstrate GTA's strong performance gains over existing approaches. These findings highlight the importance of agent-based frameworks for input adaptation and structured reasoning, offering a promising path to unlock LLMs' full potential in graph-based tasks. We hope GT Bench will be a useful resource and that GTA's principles will inspire future work on algorithmic reasoning over structured data.

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

## A  Task Definitions and Solution Algorithms in GT Bench

This section provides detailed definitions for the graph-theoretic tasks included in GT Bench, along with the canonical algorithms used to generate their ground-truth solutions. We begin by defining fundamental graph-theoretic concepts used throughout this section. A **graph** $G$ is represented as a pair $(V, E)$, where $V$ is a set of **vertices** (also called nodes) and $E$ is a set of **edges** connecting pairs of vertices. Graphs can be **directed** (edges have a direction from one vertex to another) or **undirected** (edges have no intrinsic direction). Edges may also have associated **weights** or **capacities**. A **path** is a sequence of vertices such that from each of its vertices there is an edge to the next vertex in the sequence. A **simple path** is a path where all vertices are distinct. A **cycle** is a path that starts and ends at the same vertex. A **simple cycle** is a cycle where all intermediate vertices are distinct.

**Connectivity.**    Definition: This task requires determining whether a path exists between two specified vertices, say $u$ and $v$, in a given graph $G = (V, E)$. Solution Algorithm: The ground truth is established by performing a Breadth-First Search (BFS) starting from vertex $u$. If vertex $v$ is visited during the traversal, the vertices are deemed connected; otherwise, they are not.

**Bipartiteness Check.**    Definition: This task aims to determine if the vertices of a graph $G = (V, E)$ can be partitioned into two disjoint and independent sets, $U$ and $W$, such that every edge in $E$ connects a vertex in $U$ to one in $W$. Solution Algorithm: A graph coloring approach using two colors is implemented via BFS. Starting from an arbitrary uncolored vertex and assigning it a color, its neighbors are assigned the other color, and this process continues. If an edge is found to connect two vertices of the same color, the graph is not bipartite; otherwise, it is.

**Minimum Cycle Length.**    Definition: For an unweighted graph $G = (V, E)$, this task is to find the length (i.e., the number of edges) of the shortest simple cycle. Solution Algorithm: The length of the shortest cycle is found by iterating through each vertex $s \in V$. For each $s$, a BFS is initiated. If the BFS encounters a previously visited vertex $t$ (other than the immediate parent of the current vertex in the BFS tree for $s$), a cycle is found. The length of this cycle is $\text{depth}(s_{\text{current}}) + \text{depth}(t) + 1$. The minimum such length over all starting vertices $s$ and all detected cycles forms the minimum cycle length.

**Maximum Clique Size.**    Definition: A clique in an undirected graph $G = (V, E)$ is a subset of vertices such that every two distinct vertices in the clique are adjacent. This task requires finding the number of vertices in the largest such subset. Solution Algorithm: Given that finding the maximum clique is NP-hard, a backtracking-based brute-force search algorithm is employed. This algorithm systematically explores all possible subsets of vertices, checking if each forms a clique, and keeps track of the largest clique found.

**Maximum Independent Set Size.**    Definition: An independent set in an undirected graph $G = (V, E)$ is a subset of vertices such that no two vertices in the subset are adjacent. This task asks for the size of the largest such set. Solution Algorithm: This problem is NP-hard. Similar to the maximum clique problem, a backtracking-based brute-force search algorithm is used. It explores all subsets of vertices, verifies the independent set property, and records the size of the largest valid independent set encountered.

**Eulerian Path.**    Definition: An Eulerian path in a graph is a trail that visits every edge exactly once. This task determines whether such a path exists. Solution Algorithm: For an undirected graph $G$, an Eulerian path exists if and only if $G$ is connected (considering only non-isolated vertices) and has exactly zero or two vertices of odd degree. Connectivity is checked using BFS. The degrees of all vertices are computed, and these conditions are verified. For a directed graph, similar conditions regarding in-degrees and out-degrees and strong connectivity are checked.

**Eulerian Circuit.**    Definition: An Eulerian circuit is an Eulerian path that starts and ends on the same vertex. This task determines if such a circuit exists. Solution Algorithm: For an undirected graph $G$, an Eulerian circuit exists if and only if $G$ is connected (verified by BFS) and all vertices have an even degree. For a directed graph, it exists if and only if the graph is strongly connected and every vertex $v$ has in-degree$(v)$ = out-degree$(v)$.

**Hamiltonian Path.** Definition: A Hamiltonian path is a path in an undirected or directed graph that visits each vertex exactly once. This task is to determine if such a path exists. Solution Algorithm: This problem is NP-complete. A backtracking-based brute-force search algorithm is employed. The algorithm attempts to build a path vertex by vertex, ensuring each vertex is visited exactly once, exploring all valid sequences.

**Hamiltonian Circuit.** Definition: A Hamiltonian circuit is a Hamiltonian path that is a cycle. Solution Algorithm: This problem is NP-complete. A backtracking-based brute-force search, similar to that for Hamiltonian paths, is used, with the additional constraint that the path must form a cycle by connecting the last vertex in the path back to the first.

**Biconnected Components Count.** Definition: A biconnected component of a graph is a maximal biconnected subgraph. A graph is biconnected if it remains connected if any single vertex is removed. This task counts the number of biconnected components. Solution Algorithm: The ground truth is found using a Depth-First Search (DFS)-based algorithm that identifies articulation points and bridges. Edges are then grouped into biconnected components based on these findings.

**Bridge Count.** Definition: A bridge in an undirected graph is an edge whose removal increases the number of connected components. This task requires counting the number of such bridges. Solution Algorithm: A DFS-based algorithm is used. During the DFS, discovery times ($disc[u]$) and low-link values ($low[u]$) are maintained. An edge $(u, v)$ where $v$ is a child of $u$ in the DFS tree is a bridge if $low[v] > disc[u]$.

**Triangle Count.** Definition: This task is to count the number of simple cycles of length 3 (triangles) in an undirected graph. Solution Algorithm: The number of triangles is computed by iterating through all unique triples of vertices $(u, v, w)$ and checking if edges $(u, v)$, $(v, w)$, and $(w, u)$ all exist in the graph

**Cycle Count.** Definition: This task requires counting the total number of simple cycles in the graph. Solution Algorithm: For the small graphs specified for this task in GT Bench (6-8 nodes), a DFS-based backtracking search is implemented to explicitly enumerate and count all elementary circuits.

**Spanning Tree Count.** Definition: This task is to count the number of distinct spanning trees in a given connected undirected graph. Solution Algorithm: Kirchhoff's Matrix Tree Theorem is applied. The Laplacian matrix $L$ of the graph is constructed ($L = D - A$, where $D$ is the degree matrix and $A$ is the adjacency matrix). Any cofactor of $L$ gives the number of spanning trees.

**Shortest Path Length.** Definition: This task involves computing the length of a shortest path between two specified vertices $s$ and $t$. Solution Algorithm: For unweighted graphs, Breadth-First Search (BFS) starting from $s$ is used. For weighted graphs with non-negative edge weights, Dijkstra's algorithm is employed.

**Minimum Spanning Tree (MST) Weight.** Definition: Given a connected, undirected, and edge-weighted graph, this task requires computing the total weight of a Minimum Spanning Tree (MST). Solution Algorithm: Kruskal's algorithm is used to find an MST. The sum of the weights of the edges in the found MST is the result.

**Second Minimum Spanning Tree (SMST) Weight.** Definition: This task is to find the total weight of a spanning tree whose weight is the smallest among all spanning trees that are different from an MST. Solution Algorithm: First, an MST $T$ is found using Kruskal's algorithm. Then, for each edge $(u, v)$ in the original graph $G$ not in $T$, adding $(u, v)$ to $T$ creates a cycle. On this cycle, the edge $(x, y)$ (from $T$, different from $(u, v)$) with the maximum weight is identified. Replacing $(x, y)$ with $(u, v)$ forms a new spanning tree $T'$. The SMST weight is the minimum weight among all such $T'$ that is strictly greater than the MST weight.

**Tree Diameter.** Definition: The diameter of a tree is the length of the longest shortest path between any pair of nodes. Solution Algorithm: This is found using two BFS traversals: 1. Start a BFS from

an arbitrary node $s$ to find the node $u$ farthest from $s$. 2. Start another BFS from $u$ to find the node $v$ farthest from $u$. The distance between $u$ and $v$ is the diameter.

**Tree Centroid.**  Definition: A centroid of a tree is a node $c$ such that removing $c$ divides the tree into components each with at most $|V|/2$ nodes. The task is to find such a centroid (smallest index if multiple exist). Solution Algorithm: A DFS traversal computes subtree sizes. A node $v$ is a centroid if the size of each subtree rooted at a child of $v$, and the size of the remaining tree if $v$ is removed, are all $\leq |V|/2$. The centroid with the smallest index is selected.

**Lowest Common Ancestor (LCA).**  Definition: Given a rooted tree (node 1 as root) and two nodes $u$ and $v$, the LCA is the deepest node that is an ancestor of both $u$ and $v$. Solution Algorithm: Depths and parent pointers are computed via DFS. To find LCA$(u, v)$: 1. Bring $u$ and $v$ to the same depth by moving the deeper node up. 2. If $u = v$, it is the LCA. 3. Else, move both $u$ and $v$ upwards simultaneously until they meet. This meeting point is the LCA.

**Tree Maximum Independent Set Size.**  Definition: For a given tree, find the size of a maximum independent set.  Solution Algorithm: Dynamic programming on trees is used.  For each node $u$, $dp[u][1]$ (size of max independent set in subtree of $u$, including $u$) and $dp[u][0]$ (size, excluding $u$) are computed in a post-order traversal.  $dp[u][1] = 1 + \sum_{c \in \text{children}(u)} dp[c][0]$ $dp[u][0] = \sum_{c \in \text{children}(u)} \max(dp[c][1], dp[c][0])$ The answer is $\max(dp[\text{root}][1], dp[\text{root}][0])$.

**Maximum Flow.**  Definition: Given a directed graph with edge capacities, a source $s$, and a sink $t$, find the maximum flow from $s$ to $t$. Solution Algorithm: The Edmonds-Karp algorithm is used. It iteratively finds augmenting paths from $s$ to $t$ in the residual graph using BFS and pushes flow along these paths until no more exist.

**Minimum Cut Capacity.**  Definition: In a flow network, an $s - t$ cut partitions vertices into $S$ (containing $s$) and $T$ (containing $t$). The cut capacity is the sum of capacities of edges from $S$ to $T$. This task asks for the minimum such capacity. Solution Algorithm: By the Max-Flow Min-Cut Theorem, this value is equal to the maximum flow from $s$ to $t$. Thus, the Edmonds-Karp algorithm is used to compute the maximum flow, which gives the minimum cut capacity.

**Minimum-Cost Maximum-Flow Value.**  Definition: In a flow network with edge capacities and costs per unit of flow, find a flow that maximizes total flow and, among such flows, minimizes total cost. The task asks for this minimum cost. Solution Algorithm: A successive shortest path algorithm using costs as edge lengths in the residual graph is employed. It repeatedly finds the shortest (minimum cost) augmenting path from $s$ to $t$ (using Bellman-Ford due to potential negative costs in residual graphs after flow augmentation) and pushes flow. This continues until no more augmenting paths exist or a pre-calculated maximum flow value is reached. The total cost is accumulated.

# B   RELATED WORKS

## B.1   LLMs AND GRAPH REASONING BENCHMARKS

Evaluating LLMs on graph-theoretic reasoning is crucial, yet existing benchmarks often under-assess multi-step algorithmic ability or conflate it with code-generation skills. Datasets such as GraphEval36K (Wu et al., 2024a) primarily evaluate coding proficiency on graph tasks rather than language-based reasoning over graph structure. Other works accept graph inputs but limit task difficulty or breadth: NLGraph (Wang et al., 2023) covers eight relatively simple tasks, while GPT4Graph (Guo et al., 2023) emphasizes semantic understanding rather than algorithmic graph reasoning. GraCoRe (Yuan et al., 2024) mixes trivial tasks with extremely large graphs that are ill-suited for direct LLM reasoning within context limits. GRBench (Jin et al., 2024) focuses on QA/relational reasoning, and logic datasets like ProofWriter (Tafjord et al., 2020) test rule-based inference rather than core graph-theoretic problems. Domain-specific resources (e.g., HGB (Li et al., 2023), HiGPT (Tang et al., 2024) for heterogeneous graphs, and VisionGraph (Li et al., 2024) for image graphs) provide valuable coverage but do not center on fundamental graph algorithms.

| Feature / Dimension | GT-Bench (Ours) | GraCoRe | NLGraph |
|---|---|---|---|
| Dataset Scale (Instances) | >100,000 | ~5,000 | ~30,000 |
| Number of Task Types | 44 | 19 | 8 |
| Multiple Input Representations | Yes | No | No |
| Diverse Graph Properties | Yes | Yes | Yes |
| Dynamic Graph Scaling | Yes | Yes | No |
| Multiple Difficulty Subsets | Yes | No | Yes |
| Requires Multi-Step Reasoning | Yes (all tasks) | Partial | Partial |

Table 7: Positioning GT-Bench relative to prior graph-reasoning benchmarks. Beyond scale and task breadth, GT-Bench uniquely emphasizes robustness to *representation choice* and evaluates language-based, multi-step algorithmic reasoning without code execution.

A closely related thread studies how *representation* affects LLM performance. "Talk Like a Graph" (Fatemi et al., 2023) observed that input format matters, but it did not systematically analyze representation effectiveness across diverse task types and graph structures, nor did it provide a framework to *exploit* representation sensitivity to improve problem solving.

**What GT-Bench adds.** GT-Bench targets multi-step algorithmic reasoning under realistic structural and computational constraints. It spans a broad set of classical graph problems and provides multiple input modalities (natural/structured language, adjacency list/matrix) to stress-test language-based reasoning without relying on code execution or simple lookup. To make the positioning concrete, we summarize key differences with representative benchmarks:

As Table 7 shows, GT-Bench not only expands scale and task coverage, but also stresses a previously underexplored axis—*representation sensitivity*—and evaluates whether LLMs can *reason over* graphs in language.

### B.2 ATTEMPTS TO SOLVE GRAPH-THEORETIC PROBLEMS WITH LLMS

Several recent systems attempt to apply LLMs to graph tasks, but most do not directly target language-based algorithmic reasoning over graphs.

**GraphAgent** (Yu et al., 2024) is designed for node classification on text-attributed graphs. Its problem setting and interface are specialized for classification rather than multi-step algorithmic reasoning on classical graph problems; consequently it is not directly applicable to our evaluation protocol.

**GraphAgent-Reasoner** (Hu et al., 2024b) proposes a multi-agent scheme that places an LLM agent on *every node* and coordinates them via extensive inter-agent communication. While this can achieve high accuracy in certain settings, it is computationally prohibitive for medium-sized graphs due to the explosion in model calls and message passing, and is impractical for our direct language-reasoning regime.

**GraphTeam** (Li et al., 2025) orchestrates multiple agents in a workflow (Original Question → Question Agent → Search Agent → Coding Agent), with a fallback Reasoning Agent when code fails. This *code-first* pipeline is effective for analysis workflows, but it *bypasses* intrinsic language-based graph reasoning by delegating problem solving to generated code and execution. Our objective is orthogonal: evaluate and enhance an LLM's *intrinsic* algorithmic reasoning over graph structure without relying on external code execution.

Beyond these, training- or alignment-centric approaches such as **GraphLLM** (Chai et al., 2023), **GUNDAM** (Ouyang et al., 2024), **GraphThought** (Huang et al., 2025), **GraphWiz** (Chen et al., 2024), and **GCoder** (Zhang et al., 2024b) explore instruction tuning, preference alignment, or thought-generation for graph tasks. These are promising directions, yet they typically modify or fine-tune the *executor* model itself or rely on tool/code execution. In contrast, our GTA explicitly (i) keeps the executor *frozen* and (ii) compels the model to *solve by reasoning* through an adaptive representation selector and a lightweight plan–decompose–execute pipeline. This separation isolates the benefit of better *reasoning and representation choices*, rather than changes in model capacity. Our implementation and data are open-sourced to facilitate reproducibility and fair comparison.

### B.3 LLM-Based Agent Frameworks

The view of LLMs as agents that plan, decompose, and act has gained momentum. Chain-of-Thought prompting (Wei et al., 2022) shows that exposing intermediate steps improves problem solving. ReAct (Yao et al., 2023) interleaves reasoning and actions, while Toolformer (Schick et al., 2023) trains models to decide when to invoke tools. In graph settings, Graph-CoT (Jin et al., 2024) iteratively refines reasoning over graph structure; GraphAgent variants (Yu et al., 2024; Hu et al., 2024b) introduce specialized multi-agent or planning-based pipelines, often with substantial computational overhead or with an emphasis on coding/execution. More recent automated workflow methods optimize graphs of LLM calls (e.g., Zhang et al., 2024a; Wang et al., 2025b), sometimes enabling smaller models to approach larger ones via search over plans.

Despite this progress, few agent frameworks directly target *language-based* algorithmic reasoning on graph-theoretic problems without external code execution or heavy fine-tuning. Our **Graph-Theoretic Agent (GTA)** addresses this gap with a simple, low-cost pipeline that (1) adaptively selects the most effective representation for the input graph and task, then (2) decomposes and executes solutions in a few steps, all while keeping the executor frozen. This design improves problem-solving *by reasoning*, rather than by altering the underlying model or relying on code execution.

## C  Experimental Details

This section provides further details on the experimental setup, including the models utilized, configurations for baseline methods, and specific parameters for training and evaluation.

### C.1  Models

A variety of LLMs were employed in our experiments, both for direct evaluation and as components within our proposed GTA framework and baseline methods. Table 8 (reproduced below for convenience) provides a comprehensive list of these models, including their versions, originating organizations, licenses, and their roles in our study (i.e., whether used for evaluation or fine-tuning).

Table 8: Models used in our experiments along with their versions, organizations, licenses, and purposes. *Eval*: Model used for evaluation; *FT*: Model used for fine-tuning.

| Model | Version | Organization | License | Eval | FT |
|---|---|---|---|---|---|
| Phi-4 | Phi-4 | Microsoft | MIT | ✓ | ✓ |
| GPT-4o-mini | gpt-4o-mini-2024-07-18 | OpenAI | Proprietary | ✓ | |
| GPT-4o | gpt-4o-2024-08-06 | OpenAI | Proprietary | ✓ | |
| Llama-3.1-8B | Meta-Llama-3.1-8B-Instruct | Meta | Llama 3.1 Community | ✓ | |
| Llama-3.3-70B | Meta-Llama-3.3-70B-Instruct | Meta | Llama-3.3 | ✓ | |
| QwQ | QwQ-32B | Alibaba | Apache 2.0 | ✓ | |
| o3-mini | o3-mini-2025-01-31 | OpenAI | Proprietary | ✓ | |
| Deepseek-R1 | DeepSeek-R1 | DeepSeek | MIT | ✓ | |

When these models were evaluated directly (i.e., not as part of GTA or other agentic frameworks), we employed Vanilla prompting in a zero-shot setting to assess their baseline graph reasoning capabilities.

### C.2  Baselines

For all baseline methods evaluated—Vanilla prompting, Chain-of-Thought (CoT) (Wei et al., 2022), LLM-Debate (Du et al., 2023), Self-Refine (Madaan et al., 2023), ADAS (Hu et al., 2024a), AFlow (Zhang et al., 2024a), and MaAS (Zhang et al., 2025)—we standardized the executor LLM to be `Phi-4`. This was done to ensure a fair comparison of the reasoning strategies or frameworks themselves, isolating their impact from variations in the underlying model's raw capabilities. For the automated agent frameworks (ADAS, AFlow, MaAS), the respective planner or optimizer components were configured following the specifications and recommendations detailed in their original publications.

In addition, we include two graph-specific baselines for context:

**GraphTeam (Reasoning-Only).** Following Li et al. (2025), we ablate the original multi-agent pipeline to a language-only setting by disabling search/coding/tools and external code execution. The executor remains `Phi-4` for consistency with other agentic baselines, so any improvement reflects coordination/reasoning rather than tool use.

**GraphWiz-DPO.** We evaluate the released *GraphWiz-DPO* checkpoint (Chen et al., 2024) in its prescribed instruction format and report results alongside a vanilla prompt and our GTA:

| Config | GT-E (%) | GT-H (%) |
|---|---|---|
| Vanilla | 30.2 | 15.4 |
| GraphWiz-DPO | 32.5 | 16.6 |
| GTA (LLaMA 2-13B) | 50.8 | 25.7 |

We do not include the following systems as baselines due to task mismatch, reliance on code execution (which bypasses language-based reasoning), lack of readily available code, or other practical incompatibilities with our evaluation protocol:

- **GraphAgent** (Yu et al., 2024) — targets node classification on text-attributed graphs; task/interface is mismatched to classical graph-theoretic reasoning; public code/checkpoints not available for our setting.

- **GraphAgent-Reasoner** (Hu et al., 2024b) — per-node multi-agent with heavy inter-agent communication, yielding prohibitive LLM-call costs; not peer-reviewed and no reproducible full pipeline under standard inference budgets.

- **GraphThought** (Huang et al., 2025) — focuses solely on graph combinatorial optimization (a sub-scope of graph reasoning); often relies on external tool/code execution; preprint without peer-reviewed validation and no ready-to-use code for our protocol.

- **GraphLLM** (Chai et al., 2023) — modifies the executor via fine-tuning and does not operate in our language-based, multi-format graph representation setting; preprint on older backbones (e.g., LLaMA 1/2), making comparisons with current frozen-executor agents less informative.

- **GUNDAM** (Ouyang et al., 2024) — alignment/fine-tuning of the executor (methodological paradigm differs from our frozen-executor objective); preprint status and older-model baselines further limit comparability.

- **GCoder** (Zhang et al., 2024b) — leverages code/tool execution or tuned executors, bypassing direct language-based reasoning; preprint with limited artifacts, preventing a fair, language-only comparison.

Overall, these choices align with our objective: GTA improves performance *by reasoning*—via adaptive representation selection and a lightweight plan–decompose–execute pipeline—without modifying the executor or relying on code execution.

## C.3 TRAINING AND EVALUATION PARAMETERS

**GTA Component Training:** The fine-tuning of GTA's components involved specific datasets and hyperparameters. For the **Algorithm Decomposer**, the Supervised Fine-Tuning (SFT) stage utilized a dataset of 1,000 instances where decomposition steps were distilled from `GPT-4o`. Subsequently, for Direct Preference Optimization (DPO), 2,000 preference pairs (winning vs. losing decompositions) were used. It is important to note that, given the performance characteristics of the `Phi-4` model used as the base, we constrained the number of decomposed sub-tasks, $k$, for the Algorithm Decomposer to be no more than 3. Our empirical observations indicated that when $k$ exceeded 3, the computational overhead increased while the overall accuracy of GTA tended to decrease. For training the **Learned Selector**, an initial SFT phase was conducted using 1,000 data instances. This was followed by DPO, where preference pairs were generated based on the end-to-end success of the GTA pipeline with different input representations.

Common hyperparameters for all fine-tuning experiments were as follows: models were trained for 2 epochs with a learning rate of 5.0e-6. We employed a cosine learning rate scheduler with a warmup

ratio of 0.1. The per-device training batch size was set to 1, with gradient accumulation performed over 8 steps. Training was conducted using bf16 precision to optimize memory and speed. When generating data for SFT or sampling candidates for DPO, a temperature of 1.0 was used to encourage diversity. All training experiments were conducted on 2 NVIDIA A6000 GPUs over a period of 40 hours.

**Evaluation Settings:** During all evaluation phases, including direct model assessments and runs of the baseline methods and GTA, the inference temperature for the LLMs (particularly the executor component) was consistently set to 0.01. This low temperature promotes deterministic and more replicable outputs.

For direct model evaluations on **GT Bench**, each model listed in Table 8 was tested on 10,000 instances. For the evaluation of all baseline methods and our GTA framework, each method was tested on 1,000 instances from each respective dataset (GT Bench, GraCoRe, and NLGraph).

# D    EXPERIMENTAL RESULTS

## D.1    DETAILED PERFORMANCE ON GT BENCH SUBSETS

This section provides a more granular view of model performance on the **GT Bench** dataset, specifically by delineating results on its Easy and Hard subsets. The classification of problem instances into these subsets follows a defined methodology:

Certain graph problem types are categorically assigned to a difficulty level. Specifically:

- **Hard Tasks:** All instances of "Min Cost Max Flow", "Cycle Count", "Spanning Tree Count", and "Biconnected Components" are classified as Hard, irrespective of the underlying graph's structure.

- **Easy Tasks:** All instances of "Connectivity" and "Bipartite" problems are classified as Easy, regardless of the graph structure.

For all other problem types not explicitly categorized above, the difficulty is determined by the graph's structure:

- Instances involving **sparse graphs** or **tree structures** are classified as Easy.

- Instances involving **dense graphs** are classified as Hard.

Table 9 presents the accuracy of various LLMs on these defined Easy and Hard subsets of GT Bench. Performance is reported as the percentage of correctly solved instances within each category.

Table 9: Accuracy (%) of different LLMs on the Easy and Hard subsets of GT Bench. Easy and Hard subsets are defined based on task type and graph structure as detailed in subsection D.1.

| Model | Easy Task Accuracy (%) | Hard Task Accuracy (%) |
|---|---|---|
| Llama-3.1-8B | 38.13 | 19.05 |
| GPT-4o-mini | 44.15 | 27.47 |
| Phi-4 | 53.51 | 32.97 |
| Llama-3.3-70B | 58.19 | 30.40 |
| GPT-4o | 60.54 | 28.94 |
| QwQ-32B | 85.62 | 38.46 |
| DeepSeek-r1 | 96.32 | 64.10 |
| o3-mini | 98.66 | 75.09 |

The results in Table 9 highlight the performance disparity between Easy and Hard tasks for all evaluated models, underscoring the increased challenge posed by the Hard subset of GT Bench. These figures complement the aggregate performance metrics presented in the main body of the paper.

## D.2 TASK-SPECIFIC PERFORMANCE ACROSS INPUT REPRESENTATIONS

To further investigate the impact of input graph representation on LLM performance, this section details the average accuracy achieved by the evaluated models across all tasks in GT Bench, broken down by the four input modalities: Natural Language (NL), Structured Language (SL), Adjacency Matrix (AM), and Adjacency List (AL). These results are pivotal in understanding the representation sensitivity discussed in subsection 2.4 and informed the design of the heuristic input selector within the GTA framework.

Table 10 tabulates the average accuracy for each specific task and graph type combination, ordered alphabetically by task name. The accuracy figures represent the mean performance across all LLMs tested on that particular configuration. For each task-representation pair, the highest accuracy is highlighted in bold, underscoring the optimal representation choice for that specific scenario based on our empirical data.

The data presented in Table 10 quantitatively supports the observation that no single input representation is universally optimal. For instance:

- **Natural Language (NL)** performs best or jointly best for tasks like Bipartite Check on sparse graphs, Connectivity on dense graphs, Minimum Cycle on dense graphs, and Min Cost Max Flow on dense graphs. It also shows strength in some sparse graph scenarios like Bridge Counting and Maximum Clique.

- **Structured Language (SL)** is particularly effective for tree-based problems, such as Tree Centroid, Tree Diameter, and Tree LCA. It also excels in specific non-tree tasks like Minimum Spanning Tree on sparse graphs, Second MST on dense graphs, and Spanning Tree Count on dense graphs.

- **Adjacency Matrix (AM)** demonstrates superiority for several tasks, particularly those involving global property checks or dense graph structures. Notable examples include Cycle Count on sparse graphs, Eulerian Path on both sparse and dense graphs, Maximum Clique on dense graphs, and Maximum Independent Set on both sparse and dense graphs.

- **Adjacency List (AL)** is frequently the top performer, especially for pathfinding, flow-related tasks, and some connectivity problems. Examples include Bipartite on dense graphs, Connectivity on sparse graphs, Hamiltonian Path/Circuit on both sparse and dense graphs, and Shortest Path on both graph types. It also performs well for Tree Max Independent Set.

This detailed breakdown reveals complex interactions between task type, graph structure, and representation format. The variability observed is significant, with performance differences between the best and worst representations for a given task often substantial. These findings strongly motivate the adaptive representation selection strategy employed by GTA.

## D.3 SCALABILITY ANALYSIS WITH INCREASING GRAPH SIZE

A key advantage of our automated generation process for GT Bench is the flexibility to adjust graph size (number of nodes and edges), enabling the creation of customized datasets to study LLM scalability on graph reasoning tasks. To investigate the impact of graph scale on performance, we generated variants of GT Bench tasks with increasing numbers of nodes. Figure 2 illustrates the performance trends of selected models (`Phi-4`, `GPT-4o-mini`, `QwQ-32B`) and our GTA framework (using `Phi-4` as the base) as graph size increases, separately for Easy (GT-E) and Hard (GT-H) subsets.

On the GT-E subset, which includes tasks like connectivity checks, Figure 2 shows a noticeable decline in accuracy for the smaller base models, `Phi-4` and `GPT-4o-mini`, as the number of nodes increases. In contrast, the performance of GTA remains relatively stable across the tested graph sizes. This suggests that for computationally simpler graph tasks, GTA's structured approach (adaptive representation and decomposition) effectively mitigates the challenges posed by larger graphs, maintaining robust problem-solving capabilities even when the base LLM struggles.

The performance dynamics change significantly on the GT-H subset, encompassing more complex algorithmic challenges. As graph size increases, both `Phi-4` and `GPT-4o-mini` exhibit a dramatic drop in accuracy. While highly capable models like `QwQ-32B` demonstrate greater resilience,

Table 10: Average accuracy (%) of LLMs on GT Bench tasks, segmented by input representation (NL, SL, AM, AL) and graph type (adjusted based on ground truth best representation).

| Task & Graph Type | NL (%) | SL (%) | AM (%) | AL (%) | Best Repr. |
|---|---|---|---|---|---|
| Biconnected Components (Dense) | 67.61 | 59.75 | **69.83** | 69.03 | AM |
| Biconnected Components (Sparse) | 17.85 | **23.31** | 18.90 | 22.98 | SL |
| Bipartite (Dense) | 95.63 | 95.81 | **97.64** | 96.45 | AM |
| Bipartite (Sparse) | 77.02 | 83.60 | 75.91 | **89.92** | AL |
| Bridge Count (Dense) | 83.01 | 71.93 | 84.55 | **88.23** | AL |
| Bridge Count (Sparse) | **42.91** | 36.55 | 35.14 | 40.88 | NL |
| Connectivity (Dense) | 88.79 | 89.01 | **90.65** | 87.89 | AM |
| Connectivity (Sparse) | 87.33 | 87.95 | 60.52 | **89.07** | AL |
| Cycle Count (Dense) | 12.67 | 10.98 | 11.25 | **13.01** | AL |
| Cycle Count (Sparse) | 29.78 | 24.03 | **31.84** | 26.35 | AM |
| Eulerian Circuit (Dense) | 83.05 | 75.01 | **95.03** | 92.99 | AM |
| Eulerian Circuit (Sparse) | 99.75 | **100.00** | 98.98 | 99.81 | SL |
| Eulerian Path (Dense) | 71.10 | 75.05 | **96.75** | 93.88 | AM |
| Eulerian Path (Sparse) | 98.99 | 98.15 | **100.00** | 96.42 | AM |
| Hamiltonian Circuit (Dense) | 81.35 | 80.99 | 83.03 | **88.27** | AL |
| Hamiltonian Circuit (Sparse) | 83.01 | 86.18 | 82.75 | **87.34** | AL |
| Hamiltonian Path (Dense) | 84.48 | 81.89 | 84.71 | **85.65** | AL |
| Hamiltonian Path (Sparse) | 67.68 | 68.51 | 69.03 | **71.97** | AL |
| Maximum Clique (Dense) | 24.65 | 22.08 | **38.61** | 34.87 | AM |
| Maximum Clique (Sparse) | **50.63** | 43.72 | 46.88 | 43.41 | NL |
| Maximum Flow (Dense) | 15.21 | 16.95 | **24.17** | 22.93 | AM |
| Maximum Flow (Sparse) | 34.88 | 28.03 | 33.45 | **39.56** | AL |
| Maximum Independent Set (Dense) | 27.19 | 32.61 | **36.93** | 33.17 | AM |
| Maximum Independent Set (Sparse) | 36.03 | 35.75 | **40.31** | 31.48 | AM |
| Min Cost Max Flow (Dense) | 0.13 | 0.00 | **0.98** | 0.07 | AM |
| Min Cost Max Flow (Sparse) | 21.51 | 21.20 | 23.01 | **23.45** | AL |
| Minimum Cut (Dense) | 14.70 | 15.22 | **19.03** | 18.65 | AM |
| Minimum Cut (Sparse) | 40.28 | 36.88 | 39.15 | **40.67** | AL |
| Minimum Cycle (Dense) | 86.99 | 83.07 | **87.35** | 85.33 | AM |
| Minimum Cycle (Sparse) | **81.42** | 72.49 | 80.98 | 75.03 | NL |
| Minimum Spanning Tree (Dense) | 14.39 | 16.92 | 17.22 | **19.81** | AL |
| Minimum Spanning Tree (Sparse) | 29.77 | **31.85** | 24.91 | 29.79 | SL |
| Second MST (Dense) | 14.68 | **19.83** | 11.79 | 18.09 | SL |
| Second MST (Sparse) | 23.77 | **24.15** | 22.96 | 23.70 | SL |
| Shortest Path (Dense) | 49.41 | 43.73 | 51.99 | **53.17** | AL |
| Shortest Path (Sparse) | 71.92 | 69.95 | 63.97 | **72.83** | AL |
| Spanning Tree Count (Dense) | 12.11 | 16.05 | **16.47** | 12.69 | AM |
| Spanning Tree Count (Sparse) | 18.66 | 18.08 | 19.50 | **19.85** | AL |
| Tree Centroid (Tree) | 56.25 | **60.84** | 42.59 | 51.09 | SL |
| Tree Diameter (Tree) | 37.73 | **42.05** | 34.01 | 36.03 | SL |
| Tree Max Independent Set (Tree) | 33.99 | **37.01** | 24.92 | 36.58 | SL |
| Triangle Count (Dense) | 7.82 | 16.08 | **23.29** | 18.99 | AM |
| Triangle Count (Sparse) | **40.33** | 39.17 | 37.45 | 37.77 | NL |

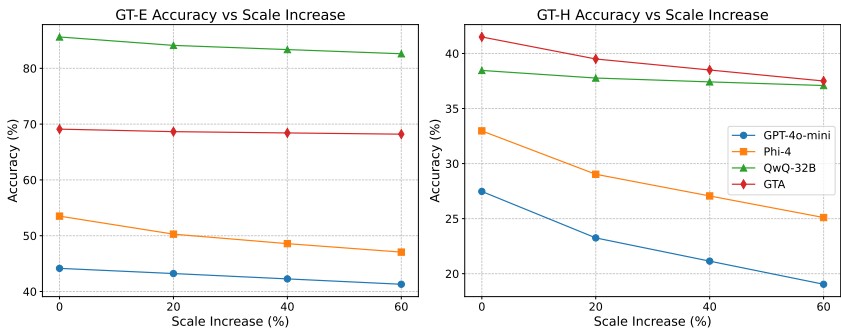

Figure 2: Performance comparison of different models (Phi-4, GPT-4o-mini, QwQ-32B) and the GTA framework on GT Bench Easy (GT-E) and Hard (GT-H) subsets as the number of graph scale increases.

maintaining strong performance even on larger hard tasks, GTA also experiences a performance decline. However, GTA's accuracy degrades much more gracefully compared to the base `Phi-4` model. Although its drop is more pronounced than that of a powerful standalone model like `QwQ-32B`, GTA consistently and significantly outperforms the vanilla `Phi-4`, indicating that the framework still provides substantial benefits for tackling complex tasks on larger graphs, even if inherent limitations of the base model eventually become more apparent.

Overall, these scalability experiments highlight GTA's ability to enhance the robustness of LLMs against increasing graph size, particularly for simpler tasks. While complex tasks on larger graphs remain challenging, GTA provides a marked improvement over using the base LLM directly.

### D.4 INFERENCE COST ANALYSIS

Beyond accuracy, the computational cost of inference is a crucial factor for the practical deployment of LLM-based reasoning frameworks. To evaluate the efficiency of GTA relative to baseline methods, we measured the token consumption and estimated the associated API costs when running each method on a representative sample of 600 instances drawn from the GT Bench dataset (combining both Easy and Hard tasks). For all methods, `Phi-4` was used as the base LLM to ensure a fair comparison of the overhead introduced by each framework. Costs are calculated based on Phi-4 pricing of $0.07 per million input tokens and $0.14 per million output tokens. Table 11 summarizes these findings.

Table 11: Inference Cost and Token Consumption (in Millions) for running 600 instances (mixed Easy/Hard) from GT Bench using Phi-4 as the base LLM. Pricing: $0.07/M input, $0.14/M output.

| Method | Input Tokens (M) | Output Tokens (M) | Estimated Cost (USD) |
|---|---|---|---|
| Phi-4 (Vanilla) | 1.75 | 0.53 | $0.20 |
| LLM-Debate | 64.51 | 19.81 | $7.40 |
| Self-Refine | 34.87 | 10.71 | $3.98 |
| AFlow | 17.44 | 5.65 | $2.12 |
| MaAS | 8.11 | 2.49 | $0.93 |
| GTA (ours) | 9.85 | 3.02 | $1.13 |

The results show significant cost differences. Methods involving multiple generation rounds or interactions, like LLM-Debate and Self-Refine, incur the highest costs, consuming substantially more tokens. AFlow also represents a notable increase over the baseline. In comparison, MaAS and our GTA framework operate with considerably lower overhead. GTA achieves its strong performance gains (as shown in subsection 4.3) at a competitive inference cost, making it a practical and efficient approach for enhancing LLM graph reasoning compared to more token-intensive methods.

### D.5 Impact of Executor Model Choice on GTA Performance

While the core GTA components (Learned Selector, Algorithm Decomposer) were fine-tuned using `Phi-4`, it is important to understand how the framework performs when a different LLM acts as the final Executor. To investigate this, we evaluated the performance of GTA using `GPT-4o-mini`, `GPT-4o`, and `DeepSeek-R1` as executors, comparing their performance within the GTA framework against their respective vanilla (direct prompting) results. The outcomes are presented in Table 12.

Table 12: Accuracy (%) comparison of different models with Vanilla prompting vs. integrated into the GTA framework as the Executor. GTA components (Selector, Decomposer) were trained using Phi-4.

| Model Configuration | GT-E (%) | GT-H (%) | GCR (%) | NLG (%) |
|---|---|---|---|---|
| GPT-4o-mini (Vanilla) | 44.2 | 27.5 | 56.2 | 70.3 |
| GPT-4o-mini (GTA Executor) | **53.1** | **34.2** | **63.5** | **76.6** |
| Phi-4 (Vanilla) | 53.5 | 33.0 | 66.8 | 80.2 |
| Phi-4 (GTA Executor) | **69.1** | **41.5** | **80.5** | **89.4** |
| GPT-4o (Vanilla) | 60.5 | 28.9 | 70.2 | 81.4 |
| GPT-4o (GTA Executor) | **68.8** | **36.9** | **78.8** | **88.2** |
| DeepSeek-R1 (Vanilla) | 96.3 | 64.1 | 97.6 | 99.0 |
| DeepSeek-R1 (GTA Executor) | 96.4 | **72.2** | 97.6 | 98.8 |

The results demonstrate that the GTA framework significantly enhances the performance of models like `GPT-4o-mini` and `GPT-4o` across all benchmarks, similar to the gains observed with `Phi-4`. For instance, integrating `GPT-4o` into GTA boosted its accuracy substantially on GT-E (from 60.5% to 68.8%) and GT-H (from 28.9% to 36.9%). This indicates that the strategic input representation and structured decomposition provided by GTA are beneficial even when the framework components are not specifically fine-tuned for that particular executor.

### D.6 SFT–DPO Training Dynamics: Bias Check and Performance Gains

We employ a two-stage pipeline for the Algorithm Decomposer and the Learned Selector: (i) **SFT** on expert demonstrations to establish a broad, modality-agnostic foundation, followed by (ii) **DPO** to directly optimize for end-to-end task success beyond imitation-style learning.

**Selector bias check.** A potential concern is whether the Heuristic-Based Selector used during DPO might bias the Decomposer toward particular input modalities. To probe this, we conduct DPO using either the Heuristic Selector or a *Random Selector* (uniform modality exposure). Results show negligible differences within standard deviation, indicating no harmful modality-specific bias:

Table 13: Effect of selector choice during DPO on Decomposer performance (mean $\pm$ std).

| Decomposer trained with | GT-E (%) | GT-H (%) | GCR (%) | NLG (%) |
|---|---|---|---|---|
| Vanilla Baseline | $53.5 \pm 0.2$ | $33.0 \pm 0.3$ | $66.8 \pm 0.1$ | $80.2 \pm 0.2$ |
| Heuristic Selector (DPO) | $\mathbf{69.1 \pm 0.4}$ | $\mathbf{41.5 \pm 0.5}$ | $\mathbf{80.2 \pm 0.3}$ | $\mathbf{89.4 \pm 0.4}$ |
| Random Selector (DPO) | $68.7 \pm 0.7$ | $40.9 \pm 0.6$ | $79.8 \pm 0.9$ | $87.0 \pm 0.5$ |

**Where DPO helps.** We apply DPO to two modules:

1) *Learned Selector.* DPO enables the selector to prefer the input representation that most likely yields a correct answer. As shown in Table 6 (main paper), the full GTA (with the Learned Selector) substantially outperforms the heuristic variant (GTA-HeuSel), confirming the value of DPO-guided representation selection.

2) *Algorithm Decomposer.* To quantify the decomposer-side gains, we compare SFT-only versus SFT+DPO. Averaged over 3 runs:

**Takeaway.** The SFT stage ensures broad coverage across tasks and modalities, while DPO provides consistent, meaningful gains by directly optimizing for task success—without introducing modality

Table 14: Decomposer ablation: SFT-only vs. SFT+DPO (mean $\pm$ std).

| Config. (Decomposer trained with...) | GT-E (%) | GT-H (%) | GCR (%) | NLG (%) |
|---|---|---|---|---|
| SFT Only | $65.1 \pm 1.1$ | $37.2 \pm 0.8$ | $71.5 \pm 0.9$ | $80.3 \pm 0.7$ |
| SFT + DPO (Ours) | $\mathbf{69.1 \pm 0.4}$ | $\mathbf{41.5 \pm 0.5}$ | $\mathbf{80.2 \pm 0.3}$ | $\mathbf{89.4 \pm 0.4}$ |

bias (Table 13) and with clear benefits over SFT-only training (Table 14). This complements the Learned Selector results (Table 6), together explaining GTA's end-to-end improvements.

## E  LIMITATIONS

**Scope of the benchmark.** Our current study concentrates on *theoretical* graph-theory tasks that are formulated independently of any specific downstream domain. Although this means we have not yet demonstrated direct transfer to citation, social, or molecular networks, we argue that establishing reliable solutions to these pure graph problems is a necessary first step toward bridging the gap between symbolic combinatorial reasoning and real-world applications.

**Graph size.** All experiments are conducted on graphs with at most 50 nodes. This decision is driven by the language-model focus of our work: when the node count grows, the number of edges in dense graphs increases quadratically, pushing both compact models such as Phi-4 and larger models like DeepSeek-R1 beyond practical context limits and greatly diminishing their utility. To facilitate future scaling studies, we publicly release an end-to-end data-generation pipeline that can instantiate graphs of any desired size; researchers can therefore regenerate the benchmark at larger scales as LLM capabilities advance.

**Statistical robustness.** Each reported score is obtained from a single inference run per model. While averaging over multiple runs would reduce stochastic variance, our computational budget is limited, and even a single full sweep already incurs substantial cost for the large-language-model cohort evaluated here. We nonetheless make all prompts, outputs, and evaluation scripts publicly available, so that subsequent work with greater resources can reproduce and extend our results.

## F  CASE STUDIES

To further illustrate the practical implications of input representation sensitivity and motivate the adaptive approach of GTA, we present several case studies. These examples showcase instances where the same LLM produces different outcomes on the identical graph problem instance, solely based on the input format used to represent the graph structure.

Our first comparison involves `GPT-4o-mini` tasked with determining connectivity on a sparse graph. As shown in Figure 3, when presented with the Natural Language (NL) description of the graph, the model incorrectly concludes the connectivity status between the specified nodes.

However, when the exact same graph instance and query are provided using the Adjacency List (AL) format (Figure 4), `GPT-4o-mini` successfully determines the correct connectivity. This suggests that the explicit listing of neighbors provided by AL facilitates the path-finding reasoning inherent in connectivity checks, which the model struggles to extract reliably from the more verbose NL description in this sparse setting.

Next, we examine `Llama-3.3-70B` attempting to find the centroid of a tree structure. Using the AL representation (Figure 5), the model fails to identify the correct centroid node(s).

In contrast, presenting the tree structure via Structured Language (SL), as depicted in Figure 6, enables `Llama-3.3-70B` to accurately compute the tree centroid. The templated, potentially hierarchical nature of the SL format seems better suited for representing tree structures in a way that facilitates reasoning about subtree sizes and properties, which are crucial for the centroid calculation. The flat neighbor listing provided by AL appears less conducive to this type of structural reasoning for tree-specific algorithms.

1296
1297
1298
1299
1300
1301
1302
1303
1304
1305
1306
1307
1308
1309
1310
1311
1312
1313
1314
1315
1316
1317
1318
1319
1320
1321
1322
1323
1324
1325
1326
1327
1328
1329
1330
1331
1332
1333
1334
1335
1336
1337
1338
1339
1340
1341
1342
1343
1344
1345
1346
1347
1348
1349

**Connectivity Sparse NL**

**Input:** Please provide the reasoning process and the final answer directly to the question. Given the graph, determine if vertex 14 and vertex 3 are connected. This is an undirected graph with 17 vertices and 17 edges. The graph contains the following connections: There is an edge between vertex 13 and vertex 11. There is an edge between vertex 12 and vertex 11. There is an edge between vertex 7 and vertex 15. There is an edge between vertex 10 and vertex 12. There is an edge between vertex 6 and vertex 12. There is an edge between vertex 13 and vertex 10. There is an edge between vertex 2 and vertex 13. There is an edge between vertex 3 and vertex 1. There is an edge between vertex 1 and vertex 12. There is an edge between vertex 9 and vertex 5. There is an edge between vertex 9 and vertex 12. There is an edge between vertex 4 and vertex 14. There is an edge between vertex 5 and vertex 15. There is an edge between vertex 8 and vertex 3. There is an edge between vertex 2 and vertex 7. There is an edge between vertex 11 and vertex 16. There is an edge between vertex 8 and vertex 4

.

**Response:** To determine if vertex 14 and vertex 3 are connected in the given undirected graph, we will analyze the connections provided. 1. List the connections: - 13 - 11 - 12 - 11 - 7 - 15 - 10 - 12 - 6 - 12 - 13 - 10 - 2 - 13 - 3 - 1 - 1 - 12 - 9 - 5 - 9 - 12 - 4 - 14 - 5 - 15 - 8 - 3 - 2 - 7 - 11 - 16 - 8 - 4 2. Identify the vertices connected to vertex 3: - Vertex 3 is connected to vertex 1 (3 - 1). - Vertex 3 is also connected to vertex 8 (8 - 3). 3. Identify the vertices connected to vertex 14: - Vertex 14 is connected to vertex 4 (4 - 14). 4. Check for a path between vertex 3 and vertex 14: - From vertex 3, we can go to vertex 1. - From vertex 1, we can go to vertex 12 (1 - 12). - From vertex 12, we can go to vertex 10 (12 - 10). - From vertex 10, we can go to vertex 13 (10 - 13). - From vertex 13, we can go to vertex 2 (13 - 2). - From vertex 2, we can go to vertex 7 (2 - 7). - From vertex 7, we can go to vertex 15 (7 - 15). - From vertex 15, we can go to vertex 5 (15 - 5). - From vertex 5, we can go to vertex 9 (5 - 9). - From vertex 9, we can go to vertex 12 (9 - 12) - but this does not help us reach vertex 14. - We cannot find a direct or indirect path from vertex 3 to vertex 14 through the connections listed. 5. Conclusion: Since there is no path connecting vertex 3 to vertex 14 through the available edges, we conclude that vertex 14 and vertex 3 are not connected. Final Answer: Vertex 14 and vertex 3 are not connected.

**Response by GPT-4o-mini** ❌

Figure 3: GPT-4o-mini incorrectly solves Connectivity (Sparse) using NL input.

## Connectivity  Sparse AL

**Input:** Please provide the reasoning process and the final answer directly to the question.  Given the graph, determine if vertex 14 and vertex 3 are connected.  This is an undirected graph with 17 vertices and 17 edges. - Adjacency List: For each vertex, lists all connected vertices.  Adjacency List: Vertex 1: 3, 12 Vertex 2: 13, 7 Vertex 3: 1, 8 Vertex 4: 14, 8 Vertex 5: 9, 15 Vertex 6: 12 Vertex 7: 15, 2 Vertex 8: 3, 4 Vertex 9: 5, 12 Vertex 10: 12, 13 Vertex 11: 13, 12, 16 Vertex 12: 11, 10, 6, 1, 9 Vertex 13: 11, 10, 2 Vertex 14: 4 Vertex 15: 7, 5 Vertex 16: 11 Vertex 17: No connections .

**Response:** To determine if vertex 14 and vertex 3 are connected in the given undirected graph, we can analyze the adjacency list provided.  1. Identify Connections:    - Vertex 14 is connected to vertex 4.    - Vertex 3 is connected to vertices 1 and 8. 2. Check for Path:    - Since vertex 14 is connected to vertex 4, we need to check if vertex 4 can connect to vertex 3 through any other vertices.    - Vertex 4 is connected to vertex 14 and vertex 8.    - Vertex 8 is connected to vertex 3 and vertex 4. 3. Establishing Connection:    - From vertex 14, we can go to vertex 4.    - From vertex 4, we can go to vertex 8.    - From vertex 8, we can go to vertex 3.  Thus, there is a path from vertex 14 to vertex 3 through the vertices 4 and 8.  Final Answer: Yes, vertex 14 and vertex 3 are connected.

### Response by GPT-4o-mini 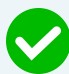

Figure 4: GPT-4o-mini correctly solves the same Connectivity (Sparse) problem using AL input.

## Tree Centroid Tree AL

**Input:** Please provide the reasoning process and the final answer directly to the question. Find the centroid of the given tree with the minimum index. A centroid is a node whose removal results in subtrees of size at most n/2. This is an undirected graph with 30 vertices and 29 edges. - Adjacency List: For each vertex, lists all connected vertices. Adjacency List: Vertex 1: 29, 7, 2, 25, 13, 24, 10, 8 Vertex 2: 1, 26 Vertex 3: 7, 28, 21, 12 Vertex 4: 28, 27 Vertex 5: 7, 22 Vertex 6: 7 Vertex 7: 1, 5, 3, 6, 30 Vertex 8: 1, 23 Vertex 9: 25, 16 Vertex 10: 1 Vertex 11: 21 Vertex 12: 3 Vertex 13: 1, 15, 14 Vertex 14: 13 Vertex 15: 13, 17 Vertex 16: 9 Vertex 17: 15 Vertex 18: 22 Vertex 19: 22 Vertex 20: 23 Vertex 21: 3, 11 Vertex 22: 5, 18, 19 Vertex 23: 8, 20 Vertex 24: 1 Vertex 25: 1, 9 Vertex 26: 2 Vertex 27: 4 Vertex 28: 3, 4 Vertex 29: 1 Vertex 30: 7

**Response: [Omit part of the Response]**Step 8: Evaluate node 1 as a potential centroid,Node 1 is connected to many other nodes, but its removal would indeed result in subtrees of varying sizes. The key is whether any of these subtrees exceed 15 nodes.,,## Step 9: Evaluate node 7 as a potential centroid,Node 7, being central and connected to nodes that themselves have multiple connections (like node 1), is a critical point. However, its removal might not result in subtrees that all have 15 nodes or fewer, especially considering the connections through node 1.,,## Step 10: Consider the tree's structure and the definition of a centroid,Given the tree's structure and the need for a centroid to have subtrees of 15 nodes or fewer upon its removal, we must carefully consider the impact of removing each highly connected node.,,## Step 11: Analyze the subtree sizes upon removal of key nodes,Upon closer inspection, removing node 1 or node 7 would indeed result in significant changes to the tree structure. However, we need to ensure that no subtree exceeds 15 nodes.,,## Step 12: Conclusion on the centroid with the minimum index,Considering the need for a centroid to divide the tree into subtrees of 15 nodes or fewer and looking for the node with the minimum index that satisfies this condition, we must identify the node that, when removed, best meets this criterion.,The final answer is: $\\boxed{7}$

## Response by Llama-3.3-70B 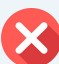

Figure 5: Llama-3.3-70B incorrectly solves Tree Centroid (Tree) using AL input.

## Tree Centroid Tree SL

**Input:** Please provide the reasoning process and the final answer directly to the question.,,Find the centroid of the given tree with the minimum index. A centroid is a node whose removal results in subtrees of size at most n/2.,,Undirected Graph with 30 vertices and 29 edges,,Edges (Format: Node Node):,1 29,1 7,7 5,1 2,7 3,1 25,3 28,1 13,13 15,7 6,3 21,5 22,22 18,28 4,15 17,1 24,7 30,25 9,1 10,9 16,21 11,1 8,3 12,4 27,22 19,13 14,2 26,8 23,23 20,

**Response: [Omit part of the Response** Step 3: Calculate the maximum size of subtrees for a centroid To find the centroid, we need to ensure that when the centroid node is removed, each resulting subtree has a size of at most n/2. Since n = 30, n/2 = 15. Therefore, we are looking for a node that, when removed, leaves no subtree with more than 15 nodes.  ## Step 4: Analyze the given edges to identify potential centroids Given the edges, we can start from any node and explore the tree to find nodes that could potentially be centroids. However, without drawing the tree or performing a detailed analysis of each node's connections, we can reason that a centroid is likely to be a node that is central and has a significant number of connections or is part of a central path in the tree.  ## Step 5: Identify the node with the minimum index that could be a centroid The problem asks for the centroid with the minimum index. This implies we should start examining nodes from the lowest index upwards. Node 1 is connected to several other nodes (29, 7, 2, 25, 13, 24, 10, 8), making it a central node with a high degree. Its removal would likely result in subtrees that are smaller than half the size of the tree, given its central position and the distribution of its connections.  ## Step 6: Confirm the choice of node 1 as a potential centroid Given the description and the connections provided, node 1 appears to be a critical central node. Its removal would split the tree into several smaller subtrees, each connected to node 1. Without a visual representation or further detailed analysis, it's reasonable to infer that node 1, due to its high connectivity and central role, could serve as a centroid that meets the condition of having subtrees of size at most n/2 when removed.
The final answer is: $\\boxed{1}$

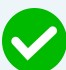

## Response by Llama-3.3-70B

Figure 6: Llama-3.3-70B correctly solves the same Tree Centroid (Tree) problem using SL input.

Our final case study features the highly capable `o3-mini` model tasked with counting cycles within a dense graph. When processing the NL description (Figure 7), the model makes an error in the cycle count.

---

## Cycle Count Dense NL

**Input:** Please provide the reasoning process and the final answer directly to the question. Count the total number of simple cycles in the given graph. This is an undirected graph with 6 vertices and 11 edges. The graph contains the following connections: There is an edge between vertex 2 and vertex 5. There is an edge between vertex 5 and vertex 3. There is an edge between vertex 4 and vertex 2. There is an edge between vertex 6 and vertex 3. There is an edge between vertex 1 and vertex 3. There is an edge between vertex 2 and vertex 1. There is an edge between vertex 1 and vertex 5. There is an edge between vertex 5 and vertex 4. There is an edge between vertex 4 and vertex 3. There is an edge between vertex 5 and vertex 6. There is an edge between vertex 3 and vertex 2.

**Response: [Omit part of the Response]** 4. 6-cycles: There is also the 6–cycle that uses all the vertices. (Again – after checking the restrictions coming from missing edges such as 1–4, 1–6, 2–6, and 4–6 – one may "list" all possibilities and find that there are 4 different 6–cycles.)

Putting it Together

Summing over all lengths we have:
- Triangles: 8
- 4–cycles: 12
- 5–cycles: 13
- 6–cycles: 4

Total = 8 + 12 + 13 + 4 = 37 simple cycles.

Final Answer
The final answer is: 37 simple cycles.

### Response by o3-mini 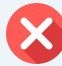

Figure 7: o3-mini incorrectly solves Cycle Count (Dense) using NL input.

Yet, when provided with the Adjacency Matrix (AM) representation for the same dense graph (Figure 8), `o3-mini` accurately calculates the number of cycles. For dense graphs, where the number of potential edges is high, the AM offers a complete and explicit view of all connections and non-connections. This comprehensive structural information appears advantageous for the complex combinatorial task of cycle counting in dense graphs, potentially mitigating ambiguities or parsing difficulties that might arise from the NL format.

1566
1567
1568
1569
1570
1571
1572
1573
1574
1575
1576
1577
1578
1579
1580
1581
1582
1583
1584
1585
1586
1587
1588
1589
1590
1591
1592
1593
1594
1595
1596
1597
1598
1599
1600
1601
1602
1603
1604
1605
1606
1607
1608
1609
1610
1611
1612
1613
1614
1615
1616
1617
1618
1619

# Cycle Count Dense AM

**Input:** Please provide the reasoning process and the final answer directly to the question. Count the total number of simple cycles in the given graph. This is an undirected graph with 6 vertices and 11 edges.
- Adjacency Matrix: 1 indicates an edge exists, 0 means no connection.
Adjacency Matrix (6×6):

```
    1  2  3  4  5  6
   ------------------------
1 | 0  1  1  0  1  0
2 | 1  0  1  1  1  0
3 | 1  1  0  1  1  1
4 | 0  1  1  0  1  0
5 | 1  1  1  1  0  1
6 | 0  0  1  0  1  0
```

**Response: [Omit part of the Response]** Step 5. Count 6-cycles (Hamiltonian cycles on all vertices). The full 6-vertex cycle, if the graph were complete, would be (6–1)!/2 = 60. However, note that vertex 6 in our graph is only adjacent to vertices 3 and 5. This forces that in any Hamiltonian cycle the two neighbors of 6 must be exactly 3 and 5. In other words, a 6-cycle "uses" 6 by "inserting" it between vertices 3 and 5. In fact, one may set up a one-to-one correspondence between a 6-cycle in the graph and a 5-cycle in the induced graph on vertices {1,2,3,4,5} that has 3 and 5 consecutive. A short enumeration shows that exactly 2 of the valid 5-cycles on {1,2,3,4,5} have 3 and 5 next to each other. (Any such 5-cycle "lifts" uniquely to a 6-cycle by replacing the edge 3–5 with the two edges 3–6 and 6–5.)
Thus there are 2 Hamiltonian (length-6) cycles.
Step 6. Add up all simple cycles.
Total cycles = triangles (3-cycles) + 4-cycles + 5-cycles + 6-cycles
= 8 + 12 + 10 + 2
= 32.

Final Answer: 32 simple cycles. 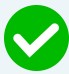

# Response by o3-mini

Figure 8: o3-mini correctly solves the same Cycle Count (Dense) problem using AM input.

These examples concretely demonstrate that even powerful LLMs exhibit significant sensitivity to how graph information is presented. The optimal representation format is not universal but rather depends intricately on the interplay between the specific task's algorithmic requirements, the underlying graph's structural properties (e.g., sparsity, density, topology like trees), and potentially the specific architecture or training of the LLM itself. This observed variability strongly underscores the necessity for adaptive representation selection, a core principle implemented within our GTA framework designed to enhance the robustness and accuracy of LLM graph reasoning.

## G  PROMPT TEMPLATES

This section details the prompt templates used for the core components of GTA: the Input Representation Selector, the Algorithm Generator, and the Algorithm Decomposer. Each prompt is designed to elicit a specific type of response from the LLM, guiding it through its designated sub-task.

---

**Prompt for Input Representation Selector**

**Your Task: Optimal Graph Representation Selection**
You will be given a graph theory problem description. Your sole responsibility is to analyze this problem and select the **single best representation format** that would make it easiest for a system to process and solve the problem algorithmically.
**Available Representation Formats:**

- NL (Natural Language: a descriptive text-based format)
- SL (Structured Language: a templated, keyword-based text format)
- AM (Adjacency Matrix: a matrix showing connections and/or weights)
- AL (Adjacency List: lists neighbors and/or weights for each vertex)

**Considerations for your selection:**

- **Problem Type:** What kind of graph question is being asked (e.g., finding paths, checking properties, calculating flows)?
- **Graph Characteristics (infer from description):**
  - Approximate size (number of nodes/edges).
  - Is it likely sparse (few edges) or dense (many edges)?
  - Are edge weights/capacities involved?
  - Is it directed or undirected?
- **Processing Efficiency:**
  - AM can be good for dense graphs or when checking non-connections is important.
  - AL is often efficient for sparse graphs and algorithms that traverse edges.
  - NL and SL might be suitable for smaller graphs or problems where the structure is simpler to describe textually, but could be harder to parse for complex algorithms.

**Input Problem:** {question}
**Instruction:** Review the input problem. Based on your analysis and the considerations above, output **only one** of the following four representation format codes: NL, SL, AM, or AL.
**Your Output (must be one of NL, SL, AM, AL):**

---

Figure 9: Prompt template for the Input Representation Selector component.

**Prompt for Algorithm Generator**

**You are an Expert Algorithm Planner for Graph Theory Problems.**
**Your Task:** You will be provided with a graph theory problem description. Your objective is to devise a **high-level strategic plan** or the **main algorithmic approach** that would be used to solve this problem.
**Crucial Instructions:**

- **DO NOT solve the problem.**
- **DO NOT derive the final answer.**
- **DO NOT write code.**

Your output should be a **conceptual outline** of the algorithm or the logical phases involved. Think of it as describing the *methodology* at a high level.
**For example, if the problem were "Find the shortest path between node A and node B":**

- A good high-level plan might be: "Utilize a breadth-first search (BFS) starting from node A, keeping track of distances, until node B is reached. Alternatively, if edges have weights, apply Dijkstra's algorithm."

- A bad (too detailed/executing) response would be: "1. Initialize distance to A as 0, all others as infinity. 2. Add A to queue. 3. While queue not empty..."

**Considerations for your plan:**

- Identify the core objective of the problem (e.g., finding a path, counting components, optimizing a value).

- What general class of graph algorithms is typically used for such a problem?

- What are the main conceptual stages of that algorithm?

**Input Problem Description:** {question}
**Instruction:** Based on the problem description, provide a concise, high-level algorithmic plan or strategy. Focus on the "what" and "why" of the approach, not the detailed "how."
**Your High-Level Algorithmic Plan:**

Figure 10: Prompt template for the Algorithm Generator component.

**Prompt for Algorithm Decomposer**

**You are an Expert Algorithm Decomposer.**
**Your Task:** You will be given an original graph theory problem description (`question`) and a high-level algorithmic plan (`plan`) designed to solve it. Your responsibility is to decompose this `plan` into a sequence of **2 or 3 highly detailed, specific, and actionable sub-steps**. These sub-steps should guide another system (an "Executor") to carry out the plan and solve the original `question`.
**Crucial Instructions:**

- **Extreme Detail Required:** Each sub-step must be very specific. Think about what data structures are needed, what values to initialize, what to iterate over, what conditions to check, what information to maintain or update at each stage.

- **Actionable by an LLM:** Phrase each sub-step as a clear instruction that an LLM could follow to perform a part of the algorithm.

- **Logical Sequence:** The sub-steps must follow a logical order that reflects the progression of the `plan`.

- **Coverage:** Together, the sub-steps should comprehensively cover the entire `plan`.

- **DO NOT solve the original problem or provide the final answer.** You are only creating the detailed execution blueprint.

**Input:**
1. **Original Problem (`question`):**

2. **High-Level Algorithmic Plan (`plan`):**

**Output Format (Strictly Adhere to This):** You MUST output the decomposed sub-steps in the following exact format. Use `### Sub-step X:` for each step.

```
### Sub-step 1:
[Provide an extremely detailed instruction for the first phase
    of the algorithm based on the plan. Be specific about:
- Initializing any necessary data structures (e.g., distance
    arrays, visited sets, queues, stacks, flow matrices).
- Setting initial values for variables or nodes (e.g., source
    node distance to 0, all others to infinity).
- The very first set of operations to perform.]
### Sub-step 2:
[Provide an extremely detailed instruction for the second phase,
    logically following Sub-step 1. Be specific about:
- Iterative processes (e.g., "While the queue is not empty..."
    or "For each neighbor of the current node...").
- Conditions for updates or state changes (e.g., "If a shorter
    path is found..." or "If the capacity is greater than zero
    ...").
- How to update data structures or values based on operations.
- What to do if a certain condition is met (e.g., target node
    reached, no more augmenting paths found).]
### Sub-step 3:
[Provide an extremely detailed instruction for the third and
    final phase, if necessary to complete the plan. Be specific
    about:
- Continuation of iterative processes.
- Termination conditions for the algorithm.
- How to derive any intermediate results needed for the final
    answer (e.g., "Sum the capacities of edges in the minimum
    cut set" or "Backtrack from the target node to reconstruct
    the path").
- Final checks or operations before the overall algorithm
    concludes.]
```

**Instruction:** Based on the provided `question` and `plan`, generate 2 or 3 highly detailed sub-steps strictly following the output format above.

Figure 11: Prompt template for the Algorithm Decomposer component.