# OpenReview forum: "GTA: Graph Theory Agent and Benchmark for Algorithmic Graph Reasoning with LLMs"
_ICLR.cc/2026/Conference — Submitted to ICLR 2026_

### Official Review · Reviewer_hj6j · 2025-10-16

**Soundness:** 3
**Presentation:** 2
**Contribution:** 2
**Rating:** 4
**Confidence:** 2

**Summary:**

The paper introduces **GT Bench**, a large-scale benchmark designed to systematically evaluate large language models on algorithmic graph reasoning tasks. It covers 44 graph problem types and over 100k instances, presented in four input formats (natural language, structured language, adjacency list, and adjacency matrix). Based on insights from GT Bench, the authors further propose **Graph Theory Agent (GTA)**—a modular framework that adaptively selects the best input representation and decomposes high-level algorithmic plans into executable substeps. Experimental results show that GTA significantly enhances LLM performance across diverse graph reasoning tasks, highlighting the crucial role of input representation and structural adaptability.

**Strengths:**

S1. The paper presents a comprehensive, well-designed benchmark that fills a major gap in evaluating LLMs’ algorithmic reasoning abilities on graph-structured data, beyond simple code generation or node-level tasks.

S2. The introduction of GTA as an adaptive, modular reasoning framework is conceptually elegant and empirically effective. The decomposition pipeline (selector → generator → decomposer → executor) is well-motivated and aligns with recent trends in agentic reasoning research.

**Weaknesses:**

W1. The paper does not address large-scale graphs that may exceed the LLM’s context window. While GT Bench focuses on medium-sized graphs suitable for in-context reasoning, real-world graph problems (e.g., social networks) often involve millions of nodes and edges that cannot fit into a single prompt. The paper lacks discussion of how GTA or LLM-based reasoning could be extended to such out-of-context or distributed graph scenarios.

W2. The paper lacks an analysis of how model size affects performance. Although the results reveal clear performance gaps among LLMs under different input representations, the paper does not examine how model scale influences these differences or whether GTA’s improvements generalize consistently across models of varying sizes.

**Questions:**

Q1. How does GTA handle or plan to handle large-scale graphs that exceed the LLM’s context window? For example, could graph partitioning, retrieval-based reasoning, or external memory mechanisms be used to extend GTA beyond in-context settings?

Q2. How does model size influence performance trends across GT Bench? Can the authors provide a detailed analysis or visualization showing how scaling (e.g., 7B, 32B...) affects reasoning robustness, representation preference, or adaptation within GTA?

Q3. Table 4 shows that different models favor different input representations. In practice, how does GTA determine the most suitable representation for a specific downstream LLM?

---

> ### Author Response · Authors · 2025-11-25
>
> W1 & Q1: On handling large-scale, out-of-context graphs.
>
> A1: We thank the reviewer for this important question regarding the scalability of our approach to very large graphs. We would like to clarify the scope and core focus of our research.
>
> 1. Complexity from Algorithmic Depth, Not Data Scale: Our work is deliberately focused on evaluating and enhancing the in-context algorithmic reasoning capabilities of LLMs. The complexity in GT Bench is not derived from the sheer scale (number of nodes) but from the combinatorial and logical depth of the tasks themselves. Problems like Hamiltonian Circuit detection or Cycle Counting are NP-hard, and **their difficulty explodes combinatorially even on the medium-sized graphs we use**. Discussing such tasks on graphs with millions of nodes is often computationally intractable even for specialized algorithms, let alone for a language-based reasoner. Therefore, extending to massive graphs is beyond the scope of what can be meaningfully considered "reasoning" in a single context.
>
> 2. GTA's "Plug-and-Play" Paradigm for In-Context Reasoning: A key contribution of our GTA framework is its **"plug-and-play" nature, operating on a frozen, unmodified executor**. This design choice makes it fundamentally incompatible with many existing large-graph techniques that require deep integration with external databases or specialized model fine-tuning. GTA's strength lies in its ability to maximize the performance of *any* given LLM for tasks that *can* fit within its context, which represents a significant and unsolved challenge in its own right.
>
> While extending LLM reasoning to out-of-context graphs is a fascinating and important future direction, our research is centered on a different, equally critical question: how can we optimize an LLM's intrinsic ability to execute complex graph algorithms through strategic representation and problem decomposition? Our work provides a robust answer to this question, laying a crucial foundation before the community moves to more complex, out-of-context scenarios.
>
> W2 & Q2: On the analysis of model size effects and performance trends.
>
> A2: We thank the reviewer for this insightful question regarding the impact of model size. While model scale is undeniably a factor, our analysis reveals that evaluating performance based on size alone is insufficient and can be misleading for complex reasoning tasks.
>
> 1. **Reasoning Capability Over Raw Scale:** We argue that specialized reasoning capabilities—such as coding proficiency and mathematical logic—are far stronger predictors of success on GT-Bench than parameter count alone. A smaller model optimized for reasoning can often outperform a larger, general-purpose model.
>
> 2. Empirical Evidence from Cross-Benchmark Comparison: To demonstrate this, we compared the performance of various models on our GT Bench (Hard subset) against their Elo ratings on the Text Arena leaderboard \[1\] across different capability dimensions (Coding, MATH, Instruction Following, Longer Query). The results are sorted by GT Bench performance:
>
> | Model | GT-H (Ours) | Coding | MATH | Instr. Follow | Longer Query |
> | :---- | :---: | :---: | :---: | :---: | :---: |
> | **Llama-3.1-8B** | 19.1% | 1258 | 1194 | 1187 | 1220 |
> | **GPT-4o-mini** | 27.5% | 1347 | 1279 | 1288 | 1322 |
> | **GPT-4o** | 28.9% | 1367 | 1308 | 1319 | 1328 |
> | **Llama-3.3-70B** | 30.4% | 1343 | 1299 | 1287 | 1309 |
> | **Phi-4** | 33.0% | 1304 | 1268 | 1239 | 1263 |
> | **QwQ-32B** | 38.5% | 1385 | 1366 | 1149 | 1176 |
> | **DeepSeek-R1** | 64.1% | **1443** | **1398** | **1390** | **1396** |
> | **o3-mini** | **75.1%** | 1413 | 1387 | 1339 | 1358 |
>
> 3. Key Findings:
>
>    * Scale is Not the Primary Driver: The most striking example is QwQ-32B. despite being a medium-sized model with lower scores in general capabilities (Instruction Following/Longer Query), it significantly outperforms the massive Llama-3.3-70B and the generally powerful GPT-4o on our graph tasks. This contradicts a simple scaling law assumption.
>    * Strong Correlation with Algorithmic Skills: The top performers on GT Bench (o3-mini, DeepSeek-R1) are also the leaders in Coding and MATH. This confirms that graph reasoning requires a specific "algorithmic intelligence" that scales with a model's formal reasoning training, rather than just its size.
>
>
>
> 4. GTA's Robustness: Furthermore, as shown in Table 12, our GTA framework provides substantial gains across this entire spectrum—boosting both the medium-sized GPT-4o and the large-scale DeepSeek-R1 (by 12.6% on GT-H). This proves that our approach is effective regardless of the underlying model size.
>
> We will include this detailed breakdown and analysis in the final version of the paper to provide a more nuanced understanding of performance factors beyond simple model scaling.

---

> > ### Author Response · Authors · 2025-11-25
> >
> > Q3: On how GTA determines the most suitable representation for a specific downstream LLM in practice.
> >
> > A3: We thank the reviewer for this question, which touches upon the core mechanism of our framework. As detailed in Section 3.1 (Input Representation Selector) of the paper, GTA employs two distinct strategies to solve exactly this problem in practice:
> >
> > Heuristic-Based Selector (General-Purpose): For a new or unknown model, we provide a rule-based selector derived from our extensive empirical analysis (Table 2). This selector maps observable graph properties (e.g., density, scale) and task types to the representation that is statistically most likely to succeed. This serves as a robust, zero-shot baseline that requires no training and generalizes well across most models.
> >
> > Learned Selector (Model-Specific Optimization): To achieve maximum performance for a specific downstream LLM (e.g., if deploying specifically for DeepSeek-R1), we use the Learned Selector. This component is a lightweight language model trained via Direct Preference Optimization using feedback from the specific executor. It learns the unique "tastes" and failure modes of that executor (e.g., "this model struggles with adjacency matrices for dense graphs"), allowing GTA to adaptively select the optimal format for each instance.
> >
> > By offering these two modes, GTA provides a flexible solution that balances immediate usability (Heuristic) with peak performance customization (Learned).
> >
> > Once again, we sincerely thank you for your review. We would be most grateful if you would consider these clarifications in your final assessment.
> >
> > [1] LMSYS Chatbot Arena Leaderboard. Retrieved from https://lmarena.ai/leaderboard/text.

---

### Official Review · Reviewer_vz5Q · 2025-10-17

**Soundness:** 3
**Presentation:** 1
**Contribution:** 3
**Rating:** 4
**Confidence:** 5

**Summary:**

This paper introduces GT Bench, a benchmark comprising 44 graph-theoretic problem types with over 100,000 instances across four input representations, designed to evaluate multi-step algorithmic reasoning capabilities of LLMs.
The authors demonstrate that LLM performance varies significantly with input representation.
 Based on these findings, they propose the Graph Theory Agent (GTA), a framework that adaptively selects optimal input representations and decomposes algorithmic solutions into sub-steps.
Experiments show GTA outperforms baselines, including prompting strategies and automated agent frameworks.

**Strengths:**

1. The paper provided a comprehensive benchmark design addressing critical gaps in previous works.
2. The GTA framework works well in the benchmark and solves the problems mentioned.
3. GTA is also cost-friendly, introducing minimal overhead.

**Weaknesses:**

1. **Bad paper organization.** Table 2 presents the best representations per task without specifying which models were evaluated or whether the results represent averages. Experimental settings (models, hyperparameters) are deferred to Section 4.1 while representation sensitivity conclusions appear in Section 2.4. Table 3 shows accuracy across representations but provides no information about which model(s) produced these results. Section 4.5 explicitly uses the Appendix to extend the paper length, leaving the main body of the paper non-self-consistent. \
I understand that there has been a lot of work done in the paper, but it is still recommended that the authors reorganize the paper to make it clearer.

2. **Scalability** All experiments restrict graphs to ≤60 nodes, which is orders of magnitude smaller than real-world graphs. Figure 2 demonstrates significant degradation even within this range for GTA.
When scaled up, methods like GraphTeam use code execution, which may offer superior scalability, yet current benchmarks may underestimate this line of work's superiority.

3. **Unfair baseline comparisons due to training data advantage.** GTA receives task-specific training exclusively on GT Bench while all baselines use only Phi-4 without GT Bench training, creating fundamental unfairness. No experiments train competitive baselines on GT Bench to isolate whether gains stem from architectural innovations or simply from task-specific fine-tuning. GraphWiz-DPO achieves only 32.5% but uses different training data and isn't adapted to GT Bench's four representations, making it unclear whether properly trained baselines would match GTA. \
**(Minor)** And it is questionable if MAS-based methods could yield stronger performance if combined with superior models, since they rely more on each agent's reasoning ability.

4. **Stronger pretrained models may obviate the need for GTA's complex framework.** Comparing Tables 4 and 5 reveals that strong pretrained models without specialized training achieve comparable performance, suggesting context capacity and general capabilities matter more than representation selection. Table 12 shows that GTA provides minimal gains for strong models.

5. **(Minor)** All main results report accuracy from single runs without standard deviations or confidence intervals.

6. As a benchmark, I would be excited if more detailed and clear analyzes is provided in the main paper. Currently, Table 2 provides raw results without further visualization, leaving the conclusions hard to interpret.
**(Minor)** The unclear results presentation also makes the current GLbench somewhat too empirical, with little insightful analysis.
One only learns that in each task, different templates outperform the others, but still don't know much about why.

**Questions:**

The paper provides no evidence that graph-theoretic reasoning tests capabilities orthogonal to existing benchmarks (MATH, GSM8K, HumanEval), leaving unclear what unique intelligence aspects GT Bench evaluates.
Will an LLM be better at reasoning, also good at graphs? Or have any interesting contradictions been found in the benchmark? (e.g. If an LLM is good at math/reasoning but fails at graphs)

---

> ### Author Response · Authors · 2025-11-25
>
> Q1: On the paper's organization, specifically the unclear data sources in tables, the early placement of conclusions, and the reliance on the appendix.
>
> A1: We sincerely thank the reviewer for their meticulous and highly valuable feedback on our paper's organization. We agree that the clarity can be improved, and we have already taken steps to address these concerns in our draft.
>
> 1. Clarified Table Provenance: We apologize for the lack of clarity in our tables. You were right to point this out. The results in Table 2 and 3 represent the average performance across all models, as detailed in Appendix D.2. We have now updated the captions in our revised manuscript to make this explicit and ensure the source of every result is unambiguous.
>
> 2. Refined Paper Narrative: We appreciate the feedback on the placement of our sensitivity analysis (Sec 2.4) before the main experiments (Sec 4). Our intent was to present the empirical motivation for GTA's design upfront. Based on your suggestion, we have carefully revised the language in Section 2.4 to frame it more clearly as a high-level, motivating analysis, ensuring a smoother logical transition to the detailed experimental validation in Section 4, thus improving the narrative flow without altering the core structure.
>
> 3. Integrating Appendix Content in Final Version: We agree that integrating key findings from the appendix would strengthen the main paper. This was a difficult compromise due to the strict page limits of the initial submission. We will fully commit to moving the most crucial analyses from the appendix into the main text in the camera-ready version, where the page limit is more accommodating.
>
> We are grateful for these concrete suggestions and are confident that these revisions make the paper much stronger and easier to follow.
>
> Q2: On the limited scalability (≤60 nodes) and the comparison to code-execution methods like GraphTeam.
>
> A2: We thank the reviewer for raising these important points on scalability and alternative methods.
>
> 1. On Graph Size and Algorithmic Complexity: Our choice of graph size was a deliberate decision to rigorously probe the limits of algorithmic reasoning.
>
>    * Complexity from Structure, Not Just Scale: Our benchmark includes a wide variety of structures, including highly dense graphs. A dense graph with 60 nodes can have nearly 1,800 edges, with its textual representation scaling quadratically (O(V²)). This complexity already pushes the context and reasoning limits of current LLMs, making direct language-based reasoning on significantly larger dense graphs an intractable problem for *any* model in the current paradigm.
>    * Focus on Reasoning, Not Throughput: Our goal is to evaluate if LLMs can perform multi-step, logical reasoning akin to executing an algorithm. The challenge in tasks like finding a Hamiltonian Circuit comes from a combinatorial explosion of possibilities, which is immense even on 60-node graphs. Our work focuses on this reasoning bottleneck, not the data ingestion challenges of massive graphs.
>
> 2. GTA Shows Improved Robustness to Scale: While performance degradation with scale is expected, our results in Figure 2 demonstrate a key strength. GTA's accuracy degrades far more gracefully than that of the base executor, proving our framework enhances robustness against increasing graph complexity.
>
> 3. Fundamental Paradigm Difference with Code-Execution Methods: As discussed in Appendix B.2, our work explores a different paradigm from tool-users like GraphTeam. Our research question centers on the pure, language-based reasoning capabilities of LLMs, without external tools like code interpreters. We keep the executor frozen and compel it to "think" through the algorithm. While code execution is a powerful, scalable solution, it bypasses this fundamental question of intrinsic reasoning. Our work is designed to evaluate and enhance this core cognitive capability, an orthogonal and equally important line of research.

---

> > ### Author Response · Authors · 2025-11-25
> >
> > Q3:Regarding the fairness of baseline comparisons and the alleged "training data advantage" for GTA.
> >
> > A3: We thank the reviewer for their concern, but we must respectfully and firmly clarify a fundamental misunderstanding of our methodology, which was detailed in Sections 3.4 and 4.1.
> >
> > 1. Strict "Frozen Executor" Paradigm: As explicitly stated in our methodology, our framework operates under a strict "frozen executor" paradigm. **The core reasoning LLM is never trained or fine-tuned on GT Bench data in any of our experiments**. This applies to GTA and all baseline methods. The performance of the core LLM is a constant; what we evaluate is the effectiveness of the external reasoning framework built around it.
> >
> > 2. Training Auxiliary Components vs. Fine-tuning the Model: The "training" mentioned in our paper is exclusively for GTA's lightweight, auxiliary modules (Selector and Decomposer). These modules learn *how to format and structure problems* for the base LLM, not the task solutions themselves. This is fundamentally different from fine-tuning the entire model on task data, which would indeed create an unfair advantage. Our comparisons are therefore a fair evaluation of different reasoning strategies applied to the same base model.
> >
> > 3. Orthogonal Approach to Fine-Tuning Methods: This distinction highlights the core difference between our approach and fine-tuning-centric methods like GraphWiz, a point we elaborate on in Appendix B.2. GraphWiz aims to improve a model's intrinsic capabilities through direct fine-tuning. Our work, in contrast, provides a lightweight, versatile framework that enhances reasoning for any given frozen LLM, without requiring costly retraining of the base model. The gains from GTA are thus attributable solely to its architectural innovations, not a data advantage.
> >
> > Q4: Concerning the necessity of GTA for stronger models and the magnitude of performance gains achieved.
> >
> > A4: We thank the reviewer for this critical analysis. We agree with the premise that a stronger base executor is the most critical factor for performance—this is a well-established principle. However, we respectfully disagree with the conclusion that this obviates the need for our framework or that our gains are "minimal."
> >
> > 1. A Strong Executor is the Foundation, GTA is the Amplifier: It is undeniable that the capabilities of the base LLM set the performance ceiling. Our framework is not intended to replace a strong model, but to amplify its capabilities. GTA acts as a "reasoning harness," ensuring that even a powerful model can apply its intelligence as effectively as possible by providing it with optimally structured problems.
> >
> > 2. Gains on Strong Models are Significant, Not Minimal: The claim that our gains are "minimal" is contradicted by our own data in Table 12\. For the powerful DeepSeek-R1, GTA boosts its performance on the highly challenging GT-H (Hard) subset from 64.1% to 72.2%—a substantial relative improvement of 12.6%. For GPT-4o, the gain on GT-H is even more dramatic, from 28.9% to 36.9%. These are significant enhancements on complex algorithmic tasks, proving that even the strongest models benefit greatly from our strategic framework.
> >
> > 3. Performance Nearing Saturation on Simpler Tasks: On easier tasks or benchmarks where models like DeepSeek-R1 already achieve near-perfect scores (e.g., 96.3% on GT-E, 99.0% on NLG), the room for improvement is naturally limited. The smaller gains in these cases are a result of a ceiling effect, as performance is already nearing saturation. However, the substantial improvements on the most difficult tasks (GT-H) are the clearest indicator of GTA's value.
> >
> > In summary, GTA acts as a synergistic partner to strong models, providing the most significant benefits where they are needed most: on complex reasoning tasks that still challenge even the best LLMs.

---

> > > ### Author Response · Authors · 2025-11-25
> > >
> > > Q5: On the desire for more detailed analysis and visualization of the benchmark results in the main paper, and the concern that the findings are currently too empirical without enough insight into "why" certain representations are better.
> > >
> > > A5: We thank the reviewer for this constructive feedback. We agree that deeper analysis and visualization would greatly enrich the paper and help readers interpret the benchmark's findings.
> > >
> > > 1. Balancing a Dual Contribution Under Page Limits: Our paper presents a dual contribution: the introduction of a large-scale benchmark (GT Bench) and a novel methodological framework (GTA). Due to the strict page limits, we had to prioritize presenting the core methodology and its evaluation. This necessitated moving much of the detailed benchmark analysis, including the insights into "why" certain representations excel into the appendix (specifically Appendix D.2).
> > >
> > > 2. Commitment to Enhanced Analysis in the Final Version: We are fully committed to improving this aspect. As the reviewer suggests, visualizations would be highly effective. In the camera-ready version, which allows for additional pages, we will add visualizations (bar charts) to the main text that summarize the key trends from our detailed analysis in Appendix D. This will provide a more intuitive and insightful overview of the relationship between task types, graph structures, and optimal representations.
> > >
> > > We appreciate the reviewer's guidance on making our benchmark's contribution more accessible and insightful, and we will ensure the final paper reflects this.
> > >
> > > Q6: On the evidence that graph-theoretic reasoning tests capabilities orthogonal to existing benchmarks.
> > >
> > > A6: This is an excellent and critical question. We have conducted a correlation analysis comparing model performance on our GT Bench (Hard subset) with their Elo ratings on Text Arena \[1\] across four dimensions: Coding, MATH, Instruction Following, and Longer Query.
> > >
> > > | Model | GT-H (Ours) | Coding | MATH | Instr. Follow | Longer Query |
> > > | :---- | :---: | :---: | :---: | :---: | :---: |
> > > | **Llama-3.1-8B** | 19.1% | 1258 | 1194 | 1187 | 1220 |
> > > | **GPT-4o-mini** | 27.5% | 1347 | 1279 | 1288 | 1322 |
> > > | **GPT-4o** | 28.9% | 1367 | 1308 | 1319 | 1328 |
> > > | **Llama-3.3-70B** | 30.4% | 1343 | 1299 | 1287 | 1309 |
> > > | **Phi-4** | 33.0% | 1304 | 1268 | 1239 | 1263 |
> > > | **QwQ-32B** | 38.5% | 1385 | 1366 | 1149 | 1176 |
> > > | **DeepSeek-R1** | 64.1% | **1443** | **1398** | **1390** | **1396** |
> > > | **o3-mini** | **75.1%** | 1413 | 1387 | 1339 | 1358 |
> > >
> > > This data reveals several striking insights that confirm the unique nature of GT Bench:
> > >
> > > 1. Orthogonality to General Capabilities: We observe a significant divergence between graph reasoning and general capabilities. The most telling example is QwQ-32B. Despite having the lowest scores in general Instruction Following (1149) and Longer Query (1176)—even lower than the 8B model—it ranks 3rd on our hard graph tasks, significantly outperforming the much larger and more generally capable GPT-4o and Llama-3.3-70B.
> > >
> > > 2. Strong Alignment with Formal Reasoning: The top performers on GT Bench (o3-mini, DeepSeek-R1, QwQ-32B) are consistently the leaders in Coding and MATH. This suggests that graph-theoretic reasoning is deeply rooted in the ability to perform precise, multi-step algorithmic execution—a capability shared with coding and mathematics but distinct from natural language understanding.
> > >
> > > 3. Debunking the "Scale is All You Need" Myth: Our benchmark highlights that model scale is not the sole determinant of reasoning success. Smaller, reasoning-optimized models (like Phi-4 and QwQ) can punch well above their weight, outperforming massive generalist models.
> > >
> > > In summary, GT Bench evaluates a specialized "algorithmic intelligence" that is not adequately captured by general benchmarks, making it a crucial tool for measuring progress in complex reasoning. We will include this comparative analysis and discussion in the revised version of our paper to clearly delineate the unique contribution of our benchmark.
> > >
> > > Once again, we sincerely thank you for your review. We would be most grateful if you would consider these clarifications in your final assessment.
> > >
> > > \[1\] LMSYS Chatbot Arena Leaderboard. Retrieved from https://lmarena.ai/leaderboard/text.

---

### Official Review · Reviewer_emXk · 2025-10-23

**Soundness:** 3
**Presentation:** 3
**Contribution:** 2
**Rating:** 4
**Confidence:** 3

**Summary:**

This paper introduces Graph Theory Bench (GT Bench) and Graph Theory Agent (GTA) to evaluate and improve LLMs abilities in algorithmic graph reasoning. GT Bench is a large-scale benchmark comprising 44 graph-theoretic tasks and over 100,000 instances presented in four input modalities, enabling systematic analysis of how representation formats and graph structures affect reasoning performance. Experiments show that LLM accuracy varies significantly across input representations, revealing strong representation sensitivity. Building on this insight, the authors propose GTA, a modular framework that adaptively selects input formats and decomposes problem-solving into structured sub-steps. Evaluations across multiple datasets demonstrate that GTA consistently outperforms existing prompting and agent-based baselines, with ablation studies confirming the contribution of its adaptive selection and decomposition modules. Overall, the paper provides a comprehensive benchmark, clear empirical insights, and an effective agentic framework that meaningfully advances the study of graph reasoning in LLMs.

**Strengths:**

- The paper offers a comprehensive empirical study. It covers 44 graph problem types and over 100,000 instances across four input modalities, includes easy/hard splits, and analyzes performance at the per-task and per-representation level. The authors also examine scalability with graph size, inference cost, and executor swaps, which gives a balanced view of where current LLMs succeed and fail.

- The methodological contribution is concrete and grounded in the findings. Building on the observed representation sensitivity, the authors design the GTA framework with an adaptive input representation selector and a plan–decompose–execute pipeline. The decomposer is trained with SFT and DPO, and ablation studies show that each module contributes to the final gains. The approach improves accuracy without relying on external code execution or modifying the base executor.

- The writing is clear and well structured. Task definitions and ground-truth algorithms are provided, prompt templates are documented, and limitations are discussed. The case studies illustrating representation effects help readers understand why certain formats are preferable for specific tasks and graph structures.

- The empirical insight on representation sensitivity is carefully substantiated. The paper systematically shows that no single input format is universally best and that performance depends on graph density, topology, and task type. This motivates the adaptive selector in GTA and adds a useful perspective for future work on language-based graph reasoning.

**Weaknesses:**

- It is unclear why not simply input all of the four graph representations together as a baseline. Maybe the result will be better?

- The paper lacks a comparison with human expert performance, even on a subset of the tasks. Including such results would help readers understand how far current LLMs have progressed in graph reasoning relative to human capabilities.

- Table 4 shows that different models prefer different graph representations, yet the training of the Selector depends on a specific model. This implies that a new Selector must be re-trained each time when the model changes, which limits the generality and practicality of the proposed approach.

- The description of the baseline in Section 4.2 is confusing. Does Vanilla prompting mean using the same workflow without additional training, or does it refer to using a single model without the agentic framework? Clarifying this distinction would make the experimental setup easier to understand.

**Questions:**

See weakness.

---

> ### Author Response · Authors · 2025-11-25
>
> Q1:On why not simply input all four graph representations together as a baseline, as this might yield better results.
>
> **A1:** We thank the reviewer for this excellent and insightful suggestion. It is a natural hypothesis that providing more information by including all representations could improve performance. We investigated this possibility for two main reasons:
>
> 1. Practical Constraints (Cost and Context Length): As the reviewer might anticipate, concatenating all four representations significantly increases the prompt length (often 3-4x longer than a single representation due to redundant information). This substantially raises inference costs and, more critically, makes the input infeasible for larger or denser graphs that would exceed the model's context window.
>
> 2. Potential for Model Confusion: Beyond practical limits, providing multiple, structurally diverse representations of the same data risks confusing the LLM. The model may struggle to prioritize the most relevant format for a given task, leading to a "cognitive overload" that can hinder its reasoning process. For example, a verbose natural language description could distract from a concise adjacency list during a pathfinding task.
>
> To empirically validate this, we conducted a new experiment on GT Bench where we provided all four representations simultaneously. The results confirm that this approach is detrimental to performance:
>
> | Model | NL | SL | AM | AL | All Reprs. |
> | :---- | :---: | :---: | :---: | :---: | :---: |
> | Llama-3.1-8B | 29.0 | 27.4 | **29.4** | 28.5 | *22.4* |
> | Phi-4 | 43.7 | **44.9** | 38.8 | 42.7 | *34.6* |
> | GPT-4o | 45.5 | 43.7 | 45.3 | **47.6** | *40.3* |
>
> As the data shows, providing all representations at once consistently degrades performance significantly compared to the best single representation for each model. This strongly supports our core thesis: **intelligent, adaptive selection of the *optimal* representation is superior to naively providing all available information**. We will add this new baseline and analysis to our final paper.
>
> Q2: Regarding the lack of comparison with human expert performance on the tasks.
>
> A2: We thank the reviewer for the suggestion. However, we respectfully argue that a direct comparison to human performance on these tasks is **fundamentally impractical** and methodologically unsound for our evaluation context.
>
> The tasks in GT Bench, especially on medium-sized graphs, require precise and extensive combinatorial calculations (e.g., counting Hamiltonian circuits, computing maximum flow) that are prohibitively difficult and error-prone for humans to perform manually. Even graph theory experts would rely on computers to solve such problems. Consequently, establishing a reliable human baseline is infeasible.
>
> This is not just a limitation of our work; it is a reflection of the nature of the tasks themselves. To our knowledge, **prior work on benchmarking LLMs for complex algorithmic reasoning on graphs has similarly not included human performance baselines**. Our focus remains on the critical and open question of how to improve the automated, algorithmic reasoning of LLMs, for which model-to-model comparison remains the most relevant and rigorous evaluation paradigm.

---

> > ### Author Response · Authors · 2025-11-25
> >
> > Q3: On the concern that the Learned Selector must be retrained for each new model, limiting the approach's generality.
> >
> > A3: We thank the reviewer for raising this important point about generality. **We would like to clarify that our framework was explicitly designed to address this challenge by offering two distinct selector options.**
> >
> > 1. Heuristic Selector Provides Strong, Zero-Shot Generality: The core of our practical contribution lies in the Heuristic-Based Selector. This component is entirely model-agnostic and requires no training whatsoever. It is derived from our foundational analysis of representation sensitivity and can be applied out-of-the-box to *any* LLM executor, providing a substantial and immediate performance boost (as shown in our GTA-HeuSel ablation). This ensures our approach is highly general and practical for a wide range of applications.
> >
> > 2. Learned Selector is for State-of-the-Art, Specialised Performance: The Learned Selector is presented as a second, more powerful option for scenarios where maximizing performance for a *specific, fixed executor* is the primary goal. While it does require retraining, this process is lightweight and targets a small selector model, not the massive executor LL. It represents an exploration of the performance ceiling, demonstrating that tailoring the representation strategy to a specific model's nuances can yield further significant gains.
> >
> > In summary, our framework offers a flexible trade-off: the Heuristic Selector for universal, plug-and-play practicality, and the Learned Selector for specialized, state-of-the-art performance. The generality of our approach is preserved through the highly effective, training-free heuristic component.
> >
> > Q4: Regarding the confusing description of the "Vanilla prompting" baseline in Section 4.2.
> >
> > A4: We sincerely apologize for the confusion and thank the reviewer for highlighting this need for clarification.
> >
> > To be precise, our "Vanilla prompting" baseline refers to using the single, base LLM executor to solve the task directly, without any of the components of our GTA agentic framework.
> >
> > The goal of this baseline is to measure the raw, unaided graph reasoning capability of the base model itself. We did mention the evaluation setting in Section 4.1 and its implementation details in Appendix C.2 , but we recognize that its definition could have been made more explicit in the main text.
> >
> > We will revise the beginning of Section 4.1 in the final version to state this definition clearly and unambiguously. We appreciate the reviewer's help in improving the clarity of our experimental setup.
> >
> > Once again, we sincerely thank you for your review. We would be most grateful if you would consider these clarifications in your final assessment.

---

### Official Review · Reviewer_GrxV · 2025-11-01

**Soundness:** 2
**Presentation:** 3
**Contribution:** 2
**Rating:** 4
**Confidence:** 3

**Summary:**

This paper introduces GT Bench, a graph reasoning benchmark with 44 problem types and four input representations, and proposes GTA, an adaptive framework combining representation selection and algorithmic decomposition. While the scale is notable, the core ideas—multi-format evaluation and stepwise reasoning—are incremental.

**Strengths:**

The introduction of GT Bench offers a diverse dataset with 44 problem types across four input modalities. Its focus on multi-step algorithmic reasoning, rather than simple graph queries or code generation, fills a gap in existing evaluations.

The empirical demonstration of LLMs' sensitivity to graph input formats is rigorous. The findings, which link optimal representation to graph structure (e.g., dense graphs with AM, trees with SL), provide actionable insights for the community, moving beyond mere observation to offer practical guidance.

**Weaknesses:**

•Incremental Conceptual Novelty: The core ideas with multi-format evaluation and algorithmic decomposition are refinements of existing concepts. The heuristic selector is simplistic, and the learned version offers only minor gains, suggesting the representation selection problem may be less complex than claimed.

•Incomplete Baseline Comparisons: The omission of direct comparisons with strong, graph-specialized models (e.g., GraphWiz) is a notable gap. It remains unclear if GTA's gains stem from its adaptive framework or if they could be matched by a model fine-tuned specifically for graph reasoning.

•Limited Scalability Demonstration: Experiments are confined to small graphs (≤50 nodes), leaving scalability to real-world scales unexplored. The work does not address the critical challenge of handling larger graphs within finite context windows, limiting the claim of evaluating "realistic" complexity.

**Questions:**

1. The representation sensitivity is a key finding. Beyond the selector, did you explore if this insight could be used to create a more robust, representation-agnostic training curriculum for LLMs to improve their inherent graph reasoning capabilities?

2. The heuristic selector performs reasonably well. In what specific scenarios or on which types of tasks does the learned selector provide the most significant advantage over the heuristic rules, and why?

3. Can GTA generalize to graphs with >100 nodes, given context limits? Providing empirical evidence and discussions could be appreciated.

---

> ### Author Response · Authors · 2025-11-25
>
> Q1: Regarding the perceived incremental novelty and the complexity of the representation selection problem.
>
> A1: We thank the reviewer for their feedback. We respectfully contend that our work introduces a novel and highly effective paradigm for LLM-based graph reasoning, and its perceived simplicity is a feature of its robustness, not a lack of novelty.
>
> 1. Pioneering Adaptive Representation Selection: To our knowledge, GTA is the **first framework to systematically leverage adaptive input representation selection as a core mechanism for enhancing LLM graph reasoning.** While multi-format evaluation and decomposition exist as general concepts, we are the first to demonstrate that *dynamically choosing the input format based on graph and task properties* is a critical, previously overlooked optimization axis. This moves beyond incremental refinement and introduces a new, powerful strategy for tackling structured data problems with LLMs. Our ablation study (Table 6\) confirms the massive impact of this novel component, showing a performance collapse without it.
>
> 2. High Efficacy with a Simple Heuristic Proves Our Insight: The strong performance of our "simplistic" heuristic selector is perhaps the most compelling evidence for the **power and validity of our core discovery**. The fact that we can achieve a substantial performance boost (e.g., GTA-HeuSel vs. GTA-NoSel) with a straightforward implementation underscores that we have identified a truly fundamental principle of LLM graph reasoning. This demonstrates the efficiency and generalizability of our approach, rather than being an argument against its novelty.
>
> 3. Learned Selector Gains are Significant, Not Minor: The gains from the learned selector are meaningful and statistically significant, especially where it matters most. On the challenging **GT-H subset, it delivers a 7.0% relative improvement (38.8% → 41.5%)**. In the complex domain of algorithmic reasoning, such an improvement is substantial.
>
> In summary, our work pioneers a new direction for LLM agents by introducing adaptive representation selection, validates this approach with a highly effective heuristic derived from our novel findings, and further advances it with a superior learned solution.
>
> Q2: Concerning the comparison with graph-specialized models (e.g., GraphWiz) and the distinction between GTA’s gains and fine-tuning.
>
> A2:  We thank the reviewer for raising this important point. We would like to respectfully clarify a misunderstanding. **We did, in fact, provide a detailed discussion and comparison with graph-specialized models, including GraphWiz, in Appendix C.2.** However, we want to first emphasize a fundamental distinction in methodology that makes direct comparison challenging.
>
> 1. Fundamentally Different Research Paradigms: Our work operates under a "frozen executor" paradigm, a crucial and practical scenario where the base LLM cannot be modified. **GTA is designed as an external, lightweight agentic framework that enhances reasoning capabilities without any fine-tuning of the executor.** In contrast, models like GraphWiz improve performance by directly fine-tuning the executor on task-specific data. These are two orthogonal approaches: GTA improves the reasoning strategy, while fine-tuning improves the reasoner's intrinsic knowledge.
> 2. Our Contribution is Orthogonal and Complementary: The primary contribution of GTA is demonstrating significant performance gains without the cost and data requirements of fine-tuning. This makes our framework highly versatile, especially for powerful, closed-source models. The gains stem directly from our adaptive architecture, which is precisely the focus of our investigation.
> 3. Performance Comparison in Appendix: As detailed in our appendix, our evaluation of a released GraphWiz checkpoint demonstrates that GTA's strategic approach can substantially outperform a model specialized through fine-tuning. This confirms that the gains from our framework are unique and not simply a substitute for fine-tuning. We kindly refer the reviewer to Appendix C.2 for the specific results and analysis.

---

> ### Author Response · Authors · 2025-11-25
>
> Q3: Concerning the limited scalability (≤50 nodes) and whether the evaluation reflects "realistic" complexity.
>
> A3: We thank the reviewer for this important question regarding scalability. Our choice of graph size up to 50 nodes was a deliberate decision designed to rigorously test the limits of algorithmic reasoning in LLMs, which we argue represents a critical and "realistic" form of complexity, distinct from raw data processing.
>
> 1. Complexity Arises from Structure, Not Just Scale: Our benchmark, GT Bench, is uniquely designed to include a wide variety of graph structures, from sparse trees to highly dense graphs. A dense graph with 50 nodes can have up to C(50, 2\) \= 1,225 edges. The textual representation of such a graph is substantial and already pushes the context limits of many LLMs. The number of edges, and thus the input length, scales quadratically with the number of nodes (O(V²)). **This exponential growth in complexity makes direct language-based reasoning on even moderately larger dense graphs an intractable problem for any current LLM**, a limitation of the current model paradigm, not our specific framework.
>
> 2. Focus on Algorithmic Reasoning, Not Data Throughput: Our goal is to evaluate whether LLMs can perform multi-step, logical reasoning akin to executing a graph algorithm. **The complexity in these tasks (e.g., finding a Hamiltonian Circuit, counting cycles) comes from the combinatorial explosion of possibilities, which is already immense in a 50-node graph. Scaling to thousands of nodes would shift the bottleneck from reasoning to simple data ingestion, which is not the focus of our work.**
>
> 3. GTA Demonstrates Superior Scalability Within Feasible Limits: While we acknowledge the size constraints, we did investigate scalability within this challenging regime. **Our analysis in Appendix D.3 shows that as graph size increases, the performance of baseline LLMs degrades sharply.** In contrast, GTA's performance degrades much more gracefully. This demonstrates that our adaptive representation and decomposition framework is a significant step toward improving the robustness of LLMs against increasing structural complexity, effectively pushing the boundary of what is feasible for language-based graph reasoning.
>
> Q4: Regarding specific scenarios where the Learned Selector significantly outperforms the heuristic rules.
>
> A4: This is a great question that gets to the heart of the learned selector's utility. The heuristic selector performs well because it captures the general, statistical trends we identified across a range of models. However, its primary limitation is that it is a "one-size-fits-all" solution. The learned selector's most significant advantage emerges in scenarios where the executor LLM deviates from these general trends—that is, when the model is an "outlier" in its processing preferences.
>
> 1. Adapting to Unique Model Architectures and "Appetites": Different LLMs, due to their unique architectures, training data, and alignment processes, can develop distinct strengths and weaknesses. For instance, a model like DeepSeek-R1, which is heavily optimized for reasoning and code, might exhibit different parsing failure modes compared to more generalist models like GPT-4o or Llama-3. The heuristic rules, derived from average performance, might not be optimal for such a specialized model. The learned selector, through DPO training, can directly learn the idiosyncratic preferences of a specific executor, choosing the representation that maximizes the end-to-end success rate *for that particular model*.
>
> 2. Handling "Edge Cases" in Graph Structure: The heuristic selector is based on clear-cut categories like "sparse" vs. "dense." It may be less effective for graphs in the ambiguous middle ground (e.g., medium density with complex community structures). In these edge cases, the optimal representation may depend on a more nuanced interplay between the task and the model's specific capabilities. The learned selector can capture these finer-grained correlations from the training data, leading to a more robust decision.
>
> In essence, while the heuristic selector provides a strong baseline based on general principles, the learned selector excels by creating a **customized, symbiotic relationship with its specific executor LLM**, offering the greatest benefits when that LLM has a unique reasoning "personality."

---

> > ### Author Response · Authors · 2025-11-25
> >
> > Q5 :On using our findings to create a representation-agnostic training curriculum for LLMs.
> > A5: We sincerely thank the reviewer for this excellent and forward-thinking question. This is a highly promising research direction that builds directly upon our work.
> >
> > **We fully agree that the insights from our systematic study of representation sensitivity are indeed foundational and could be leveraged to develop a novel training curriculum.** One could envision a multi-stage fine-tuning process where an LLM is exposed to a variety of graph representations for the same underlying problem. Such a curriculum could train the model to recognize structural invariants across different formats, thereby fostering a more abstract and robust internal understanding of graph theory concepts.
> >
> > While this is a compelling direction for improving the intrinsic capabilities of an LLM, our current work was deliberately focused on a different but equally important research question: how to enhance the performance of a pre-existing, frozen LLM through an external, agentic framework.
> >
> > We believe that our GT Bench, with its diverse tasks and parallel representations, provides the ideal foundation and necessary data infrastructure for the community to explore such training curricula in the future. We are excited by this possibility and appreciate the reviewer pointing out this valuable avenue for subsequent research.
> >
> > Once again, we sincerely thank you for your review. We would be most grateful if you would consider these clarifications in your final assessment.

---

### Meta-Review · Area_Chair_iWqk · 2026-01-06

**Summary:**

This submission introduces GT Bench, a large-scale benchmark of 44 graph-theoretic problem types with over 100,000 instances, each rendered in four graph input modalities (natural language, structured language, adjacency list, and adjacency matrix) to test multi-step algorithmic reasoning in LLMs.  Using GT Bench, the authors show that model accuracy is highly sensitive to representation and interacts with graph properties such as density and tree structure, with no single format consistently best. Motivated by this, they propose Graph Theory Agent (GTA), which selects an input representation and then applies a plan–decompose–execute pipeline, reporting accuracy gains over prompting and agent baselines on GT Bench as well as on GraCoRe and NLGraph.

Across reviews, the benchmark scope and the representation-sensitivity analysis were viewed as useful, but multiple reviewers questioned whether the methodological novelty is incremental relative to prior multi-format prompting and stepwise reasoning techniques. Reviewers also highlighted concerns about limited scalability to larger or out-of-context graphs, unclear or potentially unfair baseline comparisons, and presentation issues where important experimental details and analyses are relegated to the appendix.  The AC has carefully gone through the paper, appendices, and rebuttal, and while the work is promising, the AC is not convinced the current evidence clears the bar on novelty and evaluation rigor for ICLR, so rejection is recommended.

**Reviewer Concerns:**

The rebuttal addressed some concrete requests by adding evidence that concatenating all four representations hurts performance, clarifying that “vanilla prompting” is direct single-model inference without the agent, and pointing to an included comparison against a released GraphWiz-DPO checkpoint. It also strengthened the discussion of representation selection by explaining when the heuristic selector is intended as the general-purpose option and when the learned selector is meant for executor-specific tuning, and it reiterated the context-length motivation for limiting graphs to roughly 50–60 nodes.  However, the main outstanding issues are that the conceptual contribution still appears incremental, and the experimental story does not fully isolate whether gains come from the GTA architecture itself versus training auxiliary components on GT Bench under a setting where comparable learned baselines are not explored. Concerns about real-world scalability beyond in-context graphs and about readability (tables/protocol clarity and reliance on appendix material, plus limited statistical reporting) therefore remain unresolved for this submission.

**Reviewer Scores:**

Reviewer GrxV would likely keep the score at 4, since the rebuttal clarifies the GraphWiz comparison and selector benefits but does not eliminate their novelty and scalability reservations.  Reviewer emXk would likely raise the score slightly (to 5 or 6) because the new “all representations together” baseline and the clarifications about selector generality and the vanilla definition directly address most of their listed questions.  Reviewer vz5Q would likely stay at 4 given their high confidence and continuing concerns about fairness, scalability, and presentation, even after the authors’ clarification of the frozen-executor setting.  Reviewer hj6j would likely remain at 4, as the rebuttal responds to questions about representation choice and model-size trends but does not provide a concrete path for handling graphs that exceed the context window.

---

### Decision · Program_Chairs · 2026-01-26

Reject